# Less is More: Contrastive Retrieval Heads Improve Attention-Based Re-Ranking

## Abstract

The strong zero-shot and long-context capabilities of recent Large Language Models (LLMs) have paved the way for highly effective re-ranking systems. Attention-based re-rankers leverage attention weights from transformer heads to produce relevance scores, but not all heads are created equally: many contribute noise and redundancy, thus limiting performance. To address this, we introduce *CoRe heads*, a small set of retrieval heads identified via a contrastive scoring metric that explicitly rewards high attention heads that correlate with relevant documents, while downplaying nodes with higher attention that correlate with irrelevant documents. This relative ranking criterion isolates the most discriminative heads for re-ranking and yields a state-of-the-art list-wise re-ranker. Extensive experiments with three LLMs show that aggregated signals from CoRe heads, constituting less than $1\%$ of all heads, substantially improve re-ranking accuracy over strong baselines. We further find that CoRe heads are concentrated in middle layers, and pruning the computation of final $50\%$ of model layers preserves accuracy while significantly reducing inference time and memory usage.

## 1 Introduction

Information retrieval systems form the backbone of search engines and retrieval-augmented generation (RAG). The order of retrieved passages not only determines search relevance but also shapes the context provided to downstream generation models (Lewis et al., 2020; Gao et al., 2023). Modern retrieval systems typically adopt a two-stage pipeline: a lightweight retriever, such as BM25 (Robertson et al., 2009) or dense retrievers (Karpukhin et al., 2020), first selects a candidate set of documents, which is then refined by a re-ranker using more powerful models. Recent advances in Large Language Models (LLMs) have transformed re-ranking, delivering strong zero-shot performance (Sachan et al., 2022; Sun et al., 2023; Qin et al., 2024; Chen et al., 2025).

Recent work has extended this trend by investigating *list-wise* re-rankers, which consider the entire candidate set simultaneously and jointly produce an ordering. By exploiting cross-document attention, such models capture relative preferences across candidates, leading to rankings that align more closely with global evaluation metrics. For example, Sun et al. (2023) introduce RankGPT, which casts re-ranking as a text generation problem: the LLM is prompted to output an ordered list of document identifiers. However, this generative formulation introduces several drawbacks. Autoregressive decoding adds linear computational overhead; outputs often contain incomplete or duplicate rankings; performance is unstable due to prompt sensitivity; and the approach underuses the relevance signals already present in attention patterns. To address these challenges, Chen et al. (2025) propose an attention-based re-ranker that directly aggregates attention scores across all heads to estimate document relevance. This approach eliminates the need for text generation, reduces runtime to a single forward pass, and achieves superior accuracy compared to RankGPT. However, aggregating uniformly across all attention heads may still be suboptimal, since many heads are either redundant or capture non-informative patterns (Michel et al., 2019; Voita et al., 2019).

To better characterize the functional role of attention heads, Wu et al. (2025) identify a subset of *retrieval heads* in transformers that specialize in extracting relevant information from long contexts. Their approach tracks copy–paste operations in Needle-in-a-Haystack tasks, which constrains its applicability to narrow question-answering scenarios. Building on this line of research, Zhang et al. (2025) propose *QR heads* (query-focused retrieval heads), selected according to the absolute

attention allocated to the correct answer. While aggregated QR-head scores improve performance on downstream retrieval tasks, the selection criterion overlooks relative ranking. For instance, a head may assign high attention to the gold document yet allocate even greater weight to irrelevant documents, undermining its discriminative capacity. Since the QR-head method does not penalize such cases, it may fail to identify the heads that are most effective at distinguishing relevant from irrelevant information.

To this end, we introduce *CoRe heads* (Contrastive Retrieval heads), a subset of attention heads specialized for document re-ranking. Leveraging CoRe heads yields a state-of-the-art list-wise re-ranker that improves accuracy while simultaneously reducing computational and memory overhead, making the approach practical for real-world retrieval systems. Our contributions are threefold:

- We propose a contrastive scoring metric that identifies CoRe heads by rewarding attention directed towards relevant documents while penalizing attention to irrelevant ones. In contrast to prior retrieval-head methods that consider only absolute attention to a single document, our approach explicitly models relative ranking, resulting in stronger re-ranking accuracy.

- We demonstrate that CoRe heads identified using a small subset of the Natural Questions (NQ) dataset generalize across the BEIR benchmark (Thakur et al., 2021) and even across multi-hop tasks, cross-lingual datasets and long context settings. Attention-based re-rankers using CoRe heads consistently outperform strong baselines, achieving state-of-the-art list-wise re-ranker performance.

- We are the first to investigate layer pruning in attention-based re-rankers. We observe that top-scoring CoRe heads are primarily concentrated in the middle transformers layers. Exploiting this insight, we show that pruning most final layers incurs negligible loss in re-ranking accuracy, while reducing memory usage by $40\%$ and inference latency by $20\%$. This highlights the efficiency of CoRe heads for real-world retrieval systems.

## 2 RELATED WORK

**Zero-Shot Re-Ranking.** Zero-shot re-ranking methods with LLMs can be grouped into three main approaches: point-wise, pair-wise, and list-wise. Point-wise methods (Sachan et al., 2022; Liang et al., 2022) score each document independently, typically through logits or direct generation. These approaches are computationally efficient but often underperform due to the absence of cross-document comparison. Pair-wise methods (Qin et al., 2024) compare two documents at a time and aggregate over all pairs to form a ranking. While improving accuracy, the computational cost grows quadratically with the number of documents. List-wise methods (Ma et al., 2023; Sun et al., 2023; Chen et al., 2025) consider all documents jointly to produce a single ranked list. Although list-wise methods require long-context modeling, advances in LLMs with extended context windows (Chen et al., 2023; Jin et al., 2024a; Fu et al., 2024) have made this increasingly practical. Consequently, list-wise approaches now achieve both scalability and strong effectiveness. Our work follows this trajectory, focusing on attention-based list-wise re-ranking (Chen et al., 2025), which offers more efficiency and stability compared to generation-based methods (Ma et al., 2023; Sun et al., 2023).

**Role of Attention Heads.** A growing body of work in mechanistic interpretability examines how the activation of attention heads shapes model behavior. Michel et al. (2019) and Voita et al. (2019) show that only a small fraction of heads are necessary for translation tasks. Olsson et al. (2022) identify induction heads that capture repeated input patterns, and later extended the work to heads supporting in-context learning (Yin & Steinhardt, 2025; Ren et al., 2024). Other studies reveal specialized heads responsible for knowledge conflict (Shi et al., 2024; Jin et al., 2024b) and context distraction (Zhu et al., 2025). More recently, Wu et al. (2025) identify *retrieval heads*, which copy answer tokens from long contexts into the output, while Zhang et al. (2025) propose *query-focused retrieval heads* (QR heads) that capture query–context interactions beyond direct copying. Collectively, these studies suggest that different subsets of heads specialize in distinct computational roles, and isolating the ones relevant to retrieval remains an open challenge. In this work, we build on the notion of retrieval heads (Wu et al., 2025; Zhang et al., 2025) as the mechanisms responsible for retrieving relevant information from long-context inputs – directly aligning with the re-ranking objective.

# 3 BACKGROUND

**Attention-Based Re-Ranker.** Given a query $q$ and $k$ candidate documents $D = \{d_1, d_2, \ldots, d_k\}$ returned by a base retriever, the goal of the re-ranker is to reorder $D$ so that the documents are sorted in the order of relevance to $q$. To achieve this, Chen et al. (2025) propose In Context Re-ranking (ICR), an attention-based approach that leverages the attention signal from all attention heads of a LLM to compute fine-grained relevance scores. Utilizing the long-context capability of LLMs, ICR constructs an input prompt consisting of the list of $k$ documents followed by the query $q$:

$$``\texttt{<instruction> <}d_1\texttt{> <}d_2\texttt{> } \ldots \texttt{ <}d_k\texttt{> <instruction> <}q\texttt{>}''$$

Let $a_{j,t}^h$ be the attention score from the $t$-th token in query $q$ to the $j$-th token in document $d_i$ by an attention head $h$. For each token $j$ in document $d_i$, ICR computes the token-level relevance score with respect to the query $q$ within a head $h$ as:

$$s_{d_{i,j};q}^h = \frac{1}{|\mathcal{T}_q|} \sum_{t \in \mathcal{T}_q} a_{j,t}^h, \tag{1}$$

where $\mathcal{T}_q$ denotes the set of the query tokens.

To reduce bias relative to document length and low-information tokens (e.g., punctuation), ICR applies *contextual calibration* on the token-level score with a content-free query $q_{cf}$ (e.g. 'N/A') following Zhao et al. (2021). Specifically, the calibrated token-level score $s_{d_{i,j}}^h$ of the $j$-th token in document $d_i$ within head $h$ is calculated as:

$$s_{d_{i,j}}^h = s_{d_{i,j};q}^h - s_{d_{i,j};q_{cf}}^h, \tag{2}$$

and the document-level relevance score of document $d_i$ within head $h$ is computed as:

$$s_{d_i}^h = \sum_{t \in \mathcal{T}_{d_i}} s_{d_{i,j}}^h \tag{3}$$

where $\mathcal{T}_{d_i}$ denotes the set of the document $d_i$'s tokens. We note that the document-level scores do not involve the instruction attention scores and therefore they are not normalized post-softmax. Aggregating over all heads in the head set $H$, the final document-level scores is:

$$s_{d_i} = \sum_{h \in H} s_{d_i}^h. \tag{4}$$

**Re-Ranker with QR Heads.** As noted above, ICR computes relevance scores by aggregating attention signals across all attention heads. While comprehensive, such full-head aggregation often introduces redundancy and noise, ultimately degrading re-ranking performance. Recent work on QR heads (Zhang et al., 2025) shows that leveraging only a targeted subset of heads can improve retrieval effectiveness, suggesting that many attention heads are unnecessary for this task. Using the same notion of document-level score computed in Equation 3, a head $h$ is considered a QR head if its attention to the gold document $s_{gold}^h$ is the highest among all heads. However, the QR scoring criterion considers only the absolute attention to the gold document, without accounting for its relative ranking within each head's distribution. Because this criterion ignores how strongly a head contrasts the gold document against competing negatives, it fails to capture whether the head meaningfully separates signal from noise. This omission can lead to the selection of misleading heads. As a consequence, QR-based re-ranking exhibits inconsistent performance across datasets. For example, as shown in Figure 1, selecting the top 8 QR heads reduces accuracy on Quora for Mistral 7B and Llama 8B respectively, underperforming the ICR baseline that aggregates signals from all heads.

# 4 CONTRASTIVE RETRIEVAL HEADS

Addressing the limitations of prior approaches, we propose a contrastive head detection framework that identifies a small subset of *Contrastive Retrieval heads* (CoRe heads) specialized for document

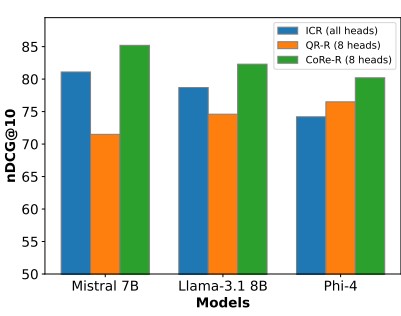

Figure 1: nDCG@10 on Quora top-40.

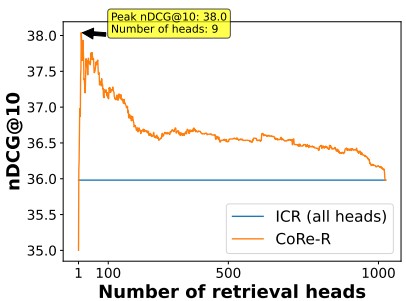

Figure 2: nDCG@10 on DBPedia top-40 with CoRe-R for Mistral 7B.

re-ranking. For clarity, we denote the default attention-based re-ranker as ICR, the QR-head re-ranker as QR-R, and the CoRe-head re-ranker as CoRe-R. Central to our method is a new scoring metric, $S_{CoRe}$, which measures a head's ability to highlight the gold document while simultaneously suppressing attention to irrelevant ones. By selecting high-quality heads using this contrastive metric, CoRe-R achieves more consistent and robust improvements in re-ranking, outperforming ICR and QR-R on Quora (Figure 1) and across the BEIR benchmark, as demonstrated in Section 5.2.

**Contrastive Scoring Metric.** Our retrieval scoring metric follows the InfoNCE loss function (Oord et al., 2018), but requires no additional training. Let $s^h_{d_{i,j};q}$ be the token-level score of token $j$ in document $d_i$ within head $h$ as defined in Equation 1. We compute the document-level score of document $d_i$ within head $h$ as:

$$s^h_{d_i} = \sum_{t \in \mathcal{T}_{d_i}} s^h_{d_{i,j};q}.$$

(5)

We note that we do not apply contextual calibration for the retrieval head scoring metric as all heads are evaluated on the same set of documents. In addition, preserving the intrinsic bias helps distinguish retrieval heads from low quality attention heads. The retrieval score of head $h$ is:

$$S_{CoRe}(h) = \frac{\exp(s^h_{pos}/t)}{\exp(s^h_{pos}/t) + \sum_i \exp(s^h_{neg,i}/t)},$$

(6)

where $t$ is a contrastive temperature hyperparameter.

The proposed contrastive scoring metric quantifies the degree to which an attention head prioritizes the gold document for a given query relative to competing irrelevant documents. A higher value of $S_{CoRe}$ indicates that the corresponding attention head more effectively discriminates the positive document from hard negative documents.

To identify CoRe heads, we compute the $S_{CoRe}$ of each attention head averaged over a set of data samples, and select the top-scoring few attention heads within each model as CoRe heads. By aggregating attention signals from these CoRe heads, the re-ranker achieves improved performance compared to aggregating over all heads. Figure 2 shows the affect of aggregating over different number of retrieval heads on the re-ranking results on the DBPedia dataset. We observe that using fewer – but higher quality – attention heads yields superior accuracy, reaching peak re-ranking accuracy with only 9 CoRe heads. Across models and datasets, we generally find that the optimal re-ranking performance is achieved with fewer than 10 CoRe heads (see Appendix C.8 for details).

**Hard Negative Data.** We compute the retrieval score $S_{CoRe}$ for all attention heads using a subset of 1000 samples from the Natural Questions (NQ) training set (Kwiatkowski et al., 2019). Each sample contains a query, a positive answer passage (gold document), and 49 hard negatives mined with the *ibm-granite/granite-embedding-30m-english* model (Awasthy et al., 2025). Hard negatives are constructed by sampling from the top 100 candidates returned by the retriever system. To further reduce false negatives, we follow de Souza P. Moreira et al. (2025) and discard any passage whose similarity to the query exceeds that of the gold passage. This procedure ensures that $S_{CoRe}$

is computed under challenging retrieval conditions, ensuring that the detected heads are more discriminative and robust for re-ranking tasks.

**Head Detection Process.** For each sample, we construct a LLM prompt consisting of the 50 documents, followed by the instruction and query (see Appendix A for details). The positive document is inserted among the first five different positions, totaling to 5000 samples per language model. We compute the $S_{CoRe}$ of each attention head on each sample, and compute the final retrieval head score by averaging across all samples. Less than top $1\%$ of all heads with the highest averaged score are selected as CoRe heads (8 heads in our experiment). This setup yields stable average retrieval scores, with the top CoRe heads consistently converging to the same subset for each model. Moreover, CoRe heads identified using a single dataset NQ generalize effectively across multiple datasets, enhancing re-ranking performance as shown later in Section 5.2. We also note that the detection process is light-weight and relatively fast, requiring less than one hour on an H100 (96GB) GPU for the models considered in this work.

**Re-Ranker with CoRe Heads.** Once CoRe heads are identified for a given language model, re-ranking is performed by aggregating attention signals exclusively from this subset. Specifically, the new relevance score $s_{d_i}^*$ in Equation 4 becomes:

$$s_{d_i} = \sum_{h \in H^*} s_{d_i}^h, \tag{7}$$

where $H^*$ denotes the set of detected CoRe heads.

## 5 EXPERIMENTS

In this section, we evaluate the effectiveness of CoRe heads across a diverse set of datasets and four different open-weight decoder models. We also analyze the effect of layer pruning, showing the proposed method achieves strong re-ranking performance without requiring the computation of all layers, significantly reducing computational overhead. Our code is available at `https://anonymous.4open.science/r/CoRe-Reranking-CCBE`.

### 5.1 SETUP

**Baselines.** We compare CoRe-R with the following zero-shot attention-based re-rankers: ICR (Chen et al., 2025) and QR-R (Zhang et al., 2025). For QR head detection, we adopt the same detection data as described in Section 4 and exclude the contextual calibration step for a fair comparison with our head detection algorithm.

**Datasets.** We evaluate our approach on the BEIR benchmark (Thakur et al., 2021) which consists of fifteen diverse datasets: TREC-COVID, NFCorpus, DBPedia-entity, SciFact, SciDocs, FiQA, NQ, FEVER, Climate-FEVER, HotpotQA, Touche, MSMARCO, Quora, ArguAna and the CQADupStack series. Among them, NQ, HotpotQA, FEVER, Climate-FEVER, and DBPedia-entity are in-domain datasets, and other datasets are out-of-domain. In addition, we evaluate on the Multilingual Long-Document Retrieval (MLDR) dataset (Chen et al., 2024) for cross-language generalization and MuSiQue (Trivedi et al., 2022) for multi-hop generalization.

**Re-ranking configuration.** For each dataset in the BEIR benchmark, we re-rank the top-$k$ documents with $k = 40, 60, 80, 100$ retrieved using the *ibm-granite/granite-embedding-30m-english* model (Awasthy et al., 2025). For the multilingual dataset MLDR, we re-rank the top-40 documents retrieved using the *ibm-granite/granite-embedding-107m-multilingual* model (Awasthy et al., 2025). Following Chen et al. (2025), we re-rank MuSiQue top-20 retrieved from ColBERT v2. Both QR-R and CoRe-R use the top 8 retrieval heads, and report performance using the nDCG@10 scores. To ensure consistency, we adopt a uniform prompt structure for all datasets; details of the prompt design are provided in Appendix A.

**Language models.** Since attention-based re-ranking requires access to all attention scores of all layers and heads, we focus on open-source LLMs. Specifically, we evaluate Mistral 7B (Jiang et al.,

Table 1: nDCG@10 on the BEIR benchmark top-40.

| Dataset | Retriever Baseline | Mistral 7B | | | Llama-3.1 8B | | | Phi-4 | | | Granite-3.2 8B | | |
|---|---|---|---|---|---|---|---|---|---|---|---|---|---|
| | | ICR | QR-R | CoRe-R | ICR | QR-R | CoRe-R | ICR | QR-R | CoRe-R | ICR | QR-R | CoRe-R |
| TREC-COVID | 63.1 | 72.1 | 73.1 | **73.8** | **76.8** | 75.7 | 75.7 | 75.2 | **76.8** | 76.0 | 70.4 | 70.3 | **71.8** |
| NFCorpus | 33.7 | 33.1 | 35.0 | **35.2** | 35.0 | 36.4 | **36.7** | 31.6 | 35.5 | **35.6** | 31.8 | 32.5 | **33.9** |
| DBPedia | 36.0 | 36.0 | 36.8 | **37.5** | 38.7 | 39.0 | **39.4** | 36.9 | 39.4 | **39.6** | 35.6 | 35.9 | **36.8** |
| SciFact | 71.3 | 71.3 | 71.9 | **74.7** | 74.7 | **75.9** | 75.1 | 71.7 | 74.2 | **74.7** | 74.6 | 74.7 | **75.2** |
| SciDocs | 22.5 | 16.6 | 17.3 | **19.5** | 18.5 | 19.7 | **20.2** | 17.8 | 21.0 | **21.1** | 18.4 | **20.3** | 20.0 |
| FiQA | 36.9 | 35.8 | 40.3 | **41.4** | 41.3 | 44.0 | **44.1** | 37.2 | 43.8 | **44.1** | 39.7 | 41.6 | **42.4** |
| NQ | 51.6 | 55.4 | 56.5 | **58.1** | 60.8 | 63.0 | **63.2** | 57.4 | 63.3 | **63.6** | 57.4 | 57.9 | **60.7** |
| FEVER | 85.5 | 87.5 | 87.1 | **88.7** | **89.2** | 87.4 | 88.4 | 89.1 | **89.6** | **89.6** | **88.0** | 87.5 | 86.2 |
| Climate-FEVER | 30.3 | 21.6 | 21.1 | **23.3** | 22.5 | **22.9** | 22.7 | 22.2 | **24.5** | 24.3 | 20.5 | **21.2** | 21.0 |
| HotpotQA | 62.9 | 71.7 | 71.6 | **73.4** | 73.7 | 73.6 | **73.8** | 73.3 | 74.2 | **74.5** | **73.6** | 72.8 | 72.6 |
| Touche | 24.0 | 23.3 | 26.4 | **26.7** | 26.1 | 26.3 | 26.4 | 21.8 | 25.4 | **25.7** | 20.5 | 24.6 | 24.8 |
| MSMARCO | 30.7 | 30.1 | **32.1** | 31.7 | 32.9 | 33.9 | **34.9** | 30.6 | 34.6 | **34.8** | 29.6 | 31.2 | **32.5** |
| Quora | 86.7 | 81.1 | 71.5 | **85.2** | 78.7 | 74.6 | **82.3** | 74.2 | 76.5 | **80.2** | **83.1** | 79.8 | 75.1 |
| ArguAna | 56.4 | 45.0 | 51.1 | **52.7** | 42.5 | 54.2 | **55.0** | 41.7 | 52.2 | **52.7** | 52.3 | 56.1 | **57.3** |
| CQADupstack | 44.3 | 38.7 | 40.8 | **41.6** | 41.7 | 43.2 | **43.9** | 39.4 | 44.0 | **44.3** | 41.4 | 42.3 | **43.4** |
| Average | 49.1 | 47.9 | 48.9 | **50.9** | 50.2 | 51.3 | **52.1** | 48.0 | 51.7 | **52.1** | 49.1 | 50.0 | **50.3** |

2023), LLama-3.1 8B (Dubey et al., 2024) and Phi-4 (Abdin et al., 2024). We additionally conduct re-ranking experiment with Granite-3.2 8B (IBM Research, 2025) which belongs to the same LLM family as the chosen retriever embedding model. Main re-ranking results for Granite-3.2 8B are shown in Table 1 and Table 2, with further results and analyses provided in Appendix C.1.

**Hyperparameters.** The temperature hyperparameter $t$ is tuned through grid search on a separate subset of the NQ train dataset. We select $t = 0.001$ for Mistral 7B and Granite-3.2 8B, and $t = 0.1$ for Llama-3.1 8B and Phi-4. These values are fixed for CoRe-R head detection prior to running the actual re-ranking experiments. Note that the effect of temperature on CoRe head identification and subsequent re-ranking performance is examined in Section 5.5.

## 5.2 Results

**Re-ranking performance on BEIR.** Table 1 reports re-ranking results on the BEIR benchmark. Across all four models, CoRe-R achieves the strongest performance, consistently outperforming both ICR and QR-R. Aggregating attention from all heads in ICR yields the weakest results, often even below the retriever baseline for Mistral 7B and Phi-4, while CoRe-R provides clear and uniform improvements over both the retriever and ICR baselines. Relative to QR-R, CoRe-R shows the largest gains on Mistral 7B, improving average nDCG@10 by +2.0 over QR-R and +3.0 over ICR, and on Llama-3.1 8B by +0.8 and +1.9 points respectively. Although QR-R remains competitive on the stronger Phi-4 model, its performance varies across datasets, whereas CoRe-R delivers consistently robust gains. In addition, we show that the gain of CoRe-R for all four LLMs is statistically significant using randomized stratified hypothesis testing (see Appendix C.3). Overall, these results demonstrate that CoRe-R reliably isolates informative attention heads across models and datasets, establishing it as a strong state-of-the-art list-wise re-ranker.

We observe that CoRe-R shows remarkable improvement on Quora compared to the baselines in Table 1. Quora is a duplicate-question retrieval task with a high proportion of hard negatives, i.e., irrelevant documents that closely resemble the gold answer. This structure aligns well with the strengths of CoRe-R: unlike ICR and QR-R, which rely on absolute attention mass and do not penalize attention to confusing negatives, CoRe-R's contrastive metric explicitly rewards heads that emphasize the positive document while suppressing near-duplicates. This suggests that contrastive head selection is particularly valuable for domains with semantically dense or near-duplicate content (e.g., FAQ retrieval, paraphrase search, code retrieval), highlighting an advantage of CoRe-R not captured by average nDCG@10 alone.

**Cross-lingual and multi-hop generalization.** Table 2 reports the nDCG@10 on the MLDR datasets across three language models. We exclude Phi-4 as the model is not intended to support multilingual use (Microsoft Research, 2024). To ensure compatibility, we evaluate only languages that are supported by all three models. With the same set of CoRe heads detected using a subset of the NQ dataset, CoRe-R shows major improvement in the re-ranking accuracy over all baselines.

Table 2: nDCG@10 on the MLDR datasets top-40.

| Dataset | Retriever Baseline | Mistral 7B | | | Llama-3.1 8B | | | Granite-3.2 8B | | |
|---|---|---|---|---|---|---|---|---|---|---|
| | | ICR | QR-R | CoRe-R | ICR | QR-R | CoRe-R | ICR | QR-R | CoRe-R |
| German | 19.9 | 23.7 | 27.2 | **28.2** | 24.9 | 27.8 | **28.6** | **28.1** | 27.1 | 27.0 |
| English | 29.5 | 23.7 | 24.6 | **28.1** | 29.1 | 29.7 | **30.2** | 28.6 | 28.3 | **29.2** |
| Spanish | 43.3 | 39.1 | 43.3 | **45.8** | 43.3 | 46.1 | **48.0** | 46.4 | 46.9 | **47.6** |
| French | 49.1 | 47.3 | 46.9 | **53.0** | 50.1 | 52.3 | **53.6** | 53.1 | 53.7 | **53.9** |
| Italian | 41.9 | 36.0 | 35.9 | **42.5** | 41.6 | 43.1 | **43.7** | 42.4 | **43.8** | **43.8** |
| Portuguese | 52.1 | 46.1 | 49.1 | **54.7** | 52.3 | 55.2 | **57.1** | 55.6 | 56.0 | **57.5** |
| Average | 39.3 | 36.0 | 37.8 | **42.1** | 40.2 | 42.4 | **43.5** | 42.4 | 42.6 | **43.2** |

For Mistral 7B, both ICR and QR-R underperform the retriever baseline on average, while CoRe-R achieves an average improvement of 2.8 points. For Llama-3.1 8B and Granite-3.2 8B, CoRe-R attains the best nDCG@10 scores on all languages with an exception of German for Granite. Figure 3 depicts the Recall@5 score on the multi-hop dataset MuSiQue, and CoRe-R continues to deliver the highest accuracy across all models with an average gain of 2.5 points over ICR and 1.2 points over QR-R (see Appendix C.5 for more results). Overall, CoRe-R demonstrates consistent gains on both cross-lingual and multi-hop tasks, underscoring the strong generalization ability of CoRe heads.

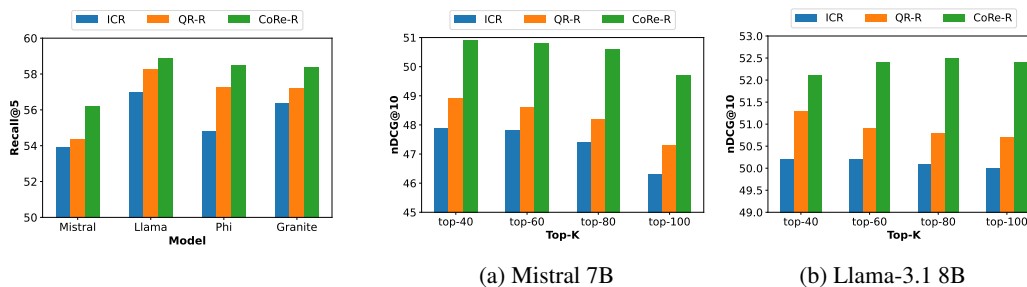

(a) Mistral 7B          (b) Llama-3.1 8B

Figure 3: Recall@5 on multi-hop dataset MuSiQue.

Figure 4: Average nDCG@10 on BEIR benchmark with different context length.

**Long context scalability.** Figure 4 reports the average accuracy on BEIR benchmark under varying context lengths (see per-dataset results in Appendix C.6). Across all settings, CoRe-R consistently outperforms both ICR and QR-R. For Mistral 7B, all methods experience noticeable degradation as the context length increases. This trend aligns with prior observations that the Mistral 7B model is not robust under long-context settings, e.g., attempts to extend its context window through long-context fine-tuning have been shown to degrade model quality (Scale AI, 2024). For Llama-3.1 8B, both baselines also show mild degradation at longer context lengths. In contrast, CoRe-R continues to improve as context grows, further widening the performance gap. This behavior highlights

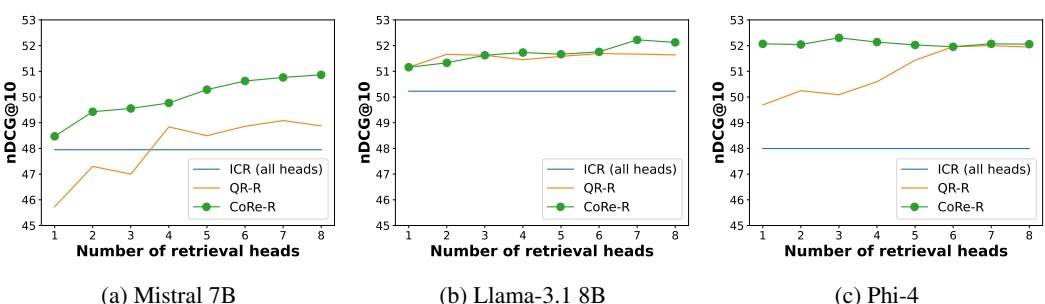

(a) Mistral 7B          (b) Llama-3.1 8B          (c) Phi-4

Figure 5: Average nDCG@10 on BEIR top-40 using different number of top retrieval heads.

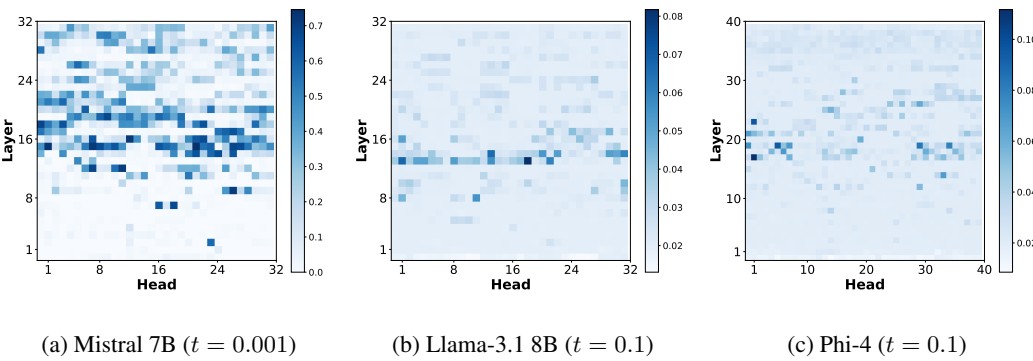

(a) Mistral 7B ($t = 0.001$)  (b) Llama-3.1 8B ($t = 0.1$)  (c) Phi-4 ($t = 0.1$)

Figure 6: Distribution of $S_{CoRe}$ for all heads in each model.

the strong scalability of CoRe heads and their ability to effectively leverage longer retrieval contexts.

**Effectiveness of few CoRe heads.** Figure 5 illustrates the re-ranking performance as a function of the number of retrieval heads across three models. Across all settings, CoRe-R consistently outperforms ICR, demonstrating the effectiveness of CoRe heads for relevance ranking across all models. For Mistral 7B, CoRe-R substantially outperforms QR-R in performance for any number of retrieval heads. With Llama-3.1 8B (Figure 5b), QR-R and CoRe-R are comparable in the re-ranking performance for six or fewer number of retrieval heads, but CoRe-R at 8 heads surpasses QR-R. Interestingly, with Phi-4 (Figure 5c), although QR-R and CoRe-R deliver close average nDCG@10 scores with top 8 retrieval heads (51.7 vs. 52.1), CoRe-R has a very strong re-ranking performance even with one single attention head (52.1), outperforming QR-R at all head levels.

## 5.3 EFFICIENT RE-RANKING VIA LAYER PRUNING

As illustrated in Figure 6, we observe that the highest scoring CoRe heads are concentrated in the middle layers across all evaluated models. We note that the distribution in Mistral 7B is more sparse due to the low temperature. Existing works have found that middle layers capture useful semantic information while late layers are primarily responsible for token generation (Clark et al., 2019; Van Aken et al., 2019). Since attention-based re-ranking does not involve text generation, the final layers are not essential for our task. We hypothesize that pruning the majority of the late layers greatly reduce memory consumption and computational cost while preserving re-ranking accuracy.

Figure 7 shows the corresponding peak GPU memory usage and Figure 8 reports the average re-ranking accuracy and latency on BEIR under different levels of layer pruning. The experiment is done on a single GPU H100 96GB. We further report the pruning experiment on the baselines retrieval heads (Wu et al., 2025) and QR heads in Appendix C.4, and the results show that their performance remains stable only under light pruning (30-40%) and begin to degrade once pruning exceeds 40%. Meanwhile, CoRe-R preserves near identical re-ranking performance under 50% pruning, reducing GPU memory usage by approximately 40% and latency by 20%. Beyond this point, CoRe-R performance begins to degrade: pruning 60% of the layers leads to a noticeable decline in the re-ranking performance, underscoring the central role of the CoRe heads in the middle layers. Nevertheless, even with

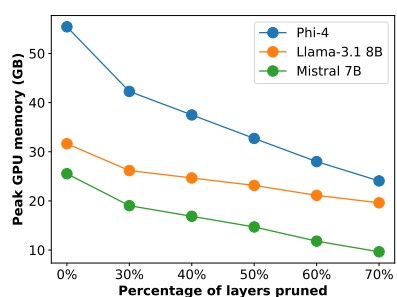

Figure 7: Peak GPU memory usage on BEIR datasets.

60% layers pruned, CoRe-R continues to outperform the ICR baseline across all models. This in-

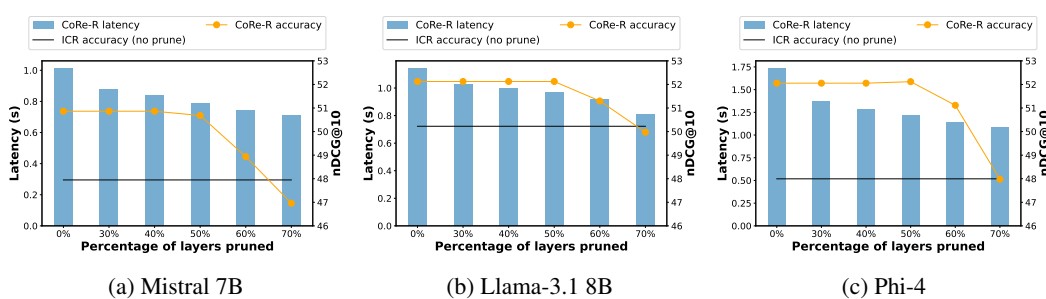

(a) Mistral 7B          (b) Llama-3.1 8B          (c) Phi-4

Figure 8: Average latency and re-ranking accuracy on BEIR benchmark.

dicates that the remaining CoRe heads in earlier layers[1] carry sufficiently strong and discriminating signals, surpassing the noisy aggregation of all heads in ICR.

## 5.4 COMPARISON OF CORE, QR, AND NIAH RETRIEVAL HEADS

In this subsection, we experiment with different set of retrieval heads, including the original retrieval heads (Wu et al., 2025) identified via copy-paste mechanism on Needle-in-a-Haystack task. We refer to these heads as *NIAH heads*. For Llama-3.1 8B, none of the top 8 CoRe heads overlaps with the top 8 NIAH heads, and only 3 overlap with the top 8 QR heads. For Mistral 7B, top 8 CoRe heads overlap with 2 QR heads and 2 NIAH heads. In addition, CoRe heads concentrate in lower layers than both QR heads and NIAH heads. We provide details of the head location in Appendix B.1.

Table 3: nDCG@10 on BEIR benchmark for Llama-3.1 8B.

| Dataset | Retriever Baseline | ICR | NIAH-R | QR-R (w/ calib) | QR-R (w/o calib) | CoRe-R |
|---|---|---|---|---|---|---|
| TREC-COVID | 63.1 | **76.8** | 73.7 | 75.2 | 75.7 | 75.7 |
| NFCorpus | 33.7 | 35.0 | 34.2 | 35.7 | 36.4 | **36.7** |
| DBPedia | 36.0 | 38.7 | 37.1 | 39.2 | 39.0 | **39.4** |
| SciFact | 71.3 | 74.7 | 74.2 | 74.4 | **75.9** | 75.1 |
| SciDocs | 22.5 | 18.5 | 19.2 | 19.2 | 19.7 | **20.2** |
| FiQA | 36.9 | 41.3 | 40.8 | 43.4 | 44.0 | **44.1** |
| NQ | 51.6 | 60.8 | 57.5 | 61.4 | 63.0 | **63.2** |
| FEVER | 85.5 | **89.2** | 88.1 | 88.5 | 87.4 | 88.4 |
| Climate-FEVER | 30.3 | 22.5 | **23.1** | 22.8 | 22.9 | 22.7 |
| HotpotQA | 62.9 | 73.7 | 72.0 | 73.5 | 73.6 | **73.8** |
| Touche | 24.0 | 26.1 | 25.5 | 26.3 | 26.3 | **26.4** |
| MSMARCO | 30.7 | 32.9 | 31.6 | 34.7 | 33.9 | **34.9** |
| Quora | 86.7 | 78.7 | 80.1 | 74.5 | 74.6 | **82.3** |
| ArguAna | 56.4 | 42.5 | 51.7 | 51.2 | 54.2 | **55.0** |
| CQADupstack | 44.3 | 41.7 | 41.4 | 43.2 | 43.2 | **43.9** |
| Average | 49.1 | 50.2 | 50.0 | 50.9 | 51.3 | **52.1** |

Table 3 compares CoRe-R with the attention-based re-rankers using the NIAH heads (NIAH-R), QR heads (without calibration) and the original QR heads (with calibration), for Llama-3.1 8B. As NIAH retrieval heads are limited to copy-paste operations, NIAH-R underperforms ICR on average, showing gains only on a few datasets such as Climate-FEVER and ArguAna, while showing worse performance on many other datasets. The original QR-R with calibration underperforms QR-R without calibration, indicating that contextual calibration is not essential for the head detection process. Meanwhile, CoRe-R shows the highest overall nDCG@10 among all baselines.

## 5.5 ABLATION STUDY

**Effect of detection dataset.** To evaluate the robustness of CoRe head selection, we repeat the detection procedure using a different dataset, MSMARCO. We found that the resulting CoRe heads are nearly identical to those detected from NQ: for both Mistral 7B and Llama-3.1 8B, 7 out of

---

[1]Specifically layers 8-12 for Mistral 7B and Llama-3.1 8B, and layers 12-16 for Phi-4

Table 4: nDCG@10 of CoRe-R on BEIR benchmark with different temperatures.

| Dataset | Retriever Baseline | Mistral 7B | | | Llama-3.1 8B | | |
|---|---|---|---|---|---|---|---|
| | | $t = 0.1$ | $t = 0.01$ | **t=0.001** | **t=0.1** | $t = 0.01$ | $t = 0.001$ |
| TREC-COVID | 63.1 | 73.6 | 73.8 | 73.8 | 75.7 | 76.3 | 76.3 |
| NFCorpus | 33.7 | 34.9 | 35.6 | 35.2 | 36.7 | 36.6 | 36.6 |
| DBPedia | 36.0 | 37.1 | 37.8 | 37.5 | 39.4 | 39.1 | 39.1 |
| SciFact | 71.3 | 72.2 | 73.0 | 74.7 | 75.1 | 75.9 | 75.9 |
| SciDocs | 22.5 | 17.1 | 18.4 | 19.5 | 20.2 | 20.0 | 20.0 |
| FiQA | 36.9 | 39.5 | 41.2 | 41.4 | 44.1 | 44.3 | 44.3 |
| NQ | 51.6 | 55.6 | 57.8 | 58.1 | 63.2 | 63.6 | 63.6 |
| FEVER | 85.5 | 86.9 | 87.7 | 88.7 | 88.4 | 88.2 | 88.2 |
| Climate-FEVER | 30.3 | 21.1 | 21.9 | 23.3 | 22.7 | 22.9 | 22.9 |
| HotpotQA | 62.9 | 71.6 | 72.4 | 73.4 | 73.8 | 73.4 | 73.4 |
| Touche | 24.0 | 26.6 | 27.1 | 26.2 | 26.4 | 26.9 | 26.9 |
| MSMARCO | 30.7 | 31.7 | 32.3 | 31.7 | 34.9 | 34.8 | 34.8 |
| Quora | 86.7 | 73.4 | 81.5 | 85.2 | 82.3 | 74.5 | 74.5 |
| ArguAna | 56.4 | 50.8 | 52.0 | 52.7 | 55.0 | 54.5 | 54.5 |
| CQADupstack | 44.3 | 40.3 | 41.4 | 41.6 | 43.9 | 43.6 | 43.6 |
| Average | 49.1 | 48.8 | 50.3 | 50.9 | 52.1 | 51.6 | 51.6 |

the 8 selected heads overlap, and the one different head comes close within the top-10 heads (see Appendix C.7). We then performed the re-ranking experiments using these MSMARCO-derived heads, and the results are reported in Appendix C.7. With 7 out of 8 heads overlapping, the re-ranking performance closely matches the results obtained using NQ, demonstrating that CoRe head selection is highly stable and does not depend heavily on the detection dataset.

**Effect of temperature.** We perform ablation study on the temperature with different values $t = 0.1, 0.01, 0.001$. Since $t$ directly controls the sharpness of the contrastive scoring metric, different temperatures can change the shape of the score distribution and may affect which heads appear in the top-ranked set. We found that the detected CoRe heads in Llama-3.1 8B are consistent across all temperature values while Mistral 7B highly depends on the temperature with 4 common heads detected out of 8 heads (see Appendix B). We observe that many heads in Mistral attend strongly to both the positive document and hard negatives, thus, higher temperatures $t = 0.1, 0.01$ were not able to capture high-quality heads in Mistral 7B.

Table 4 shows the re-ranking accuracy per dataset under different temperatures. As expected, the results for Llama are similar across all three temperature values, while Mistral demonstrates higher impact from the different detected head set. While a small amount of temperature tuning is necessary for certain models, the entire tuning process incurs minimal overhead as head detection is a relatively fast one-time process per language model. Our experiments on selected LLMs show that the entire head detection process takes less than an hour per model on an H-100 96GB GPU machine. Once selected, the same CoRe heads generalize reliably across cross-domain tasks, multi-hop retrieval, and long-context settings. This demonstrates that CoRe's performance is not sensitive to temperature in practice, and the detected heads remain robust across applications.

# 6 CONCLUSION

In this paper, we introduced *Contrastive Retrieval heads* (CoRe heads), a subset of attention heads that capture the most discriminative signals for document re-ranking. We proposed a contrastive scoring metric that identifies these heads by rewarding attention to relevant documents while penalizing focus on irrelevant ones. Across extensive experiments, we showed that aggregating attention from CoRe heads produces a state-of-the-art re-ranker, consistently surpassing prior baselines across tasks and models. Our analysis further revealed that CoRe heads cluster in the middle transformer layers, enabling an effective layer-pruning strategy that cuts inference latency and memory usage without sacrificing accuracy. Together, these results establish CoRe heads as primary carriers of relevance information for re-ranking and underscore their promise for building fast, accurate, and efficient retrieval systems.

## 7 REPRODUCIBILITY STATEMENT

- Code and Scripts: All code, scripts, and configuration files are available at `https://anonymous.4open.science/r/CoRe-Reranking-CCBE`
- Datasets: We use only publicly available datasets: Natural Questions (NQ), BEIR (15 datasets), and MLDR.
- Preprocessing: Hard negative mining and filtering are described in Section 4.
- Hyperparameters: All hyperparameters, including contrastive temperature values, are reported in Section 5.1.
- LLM Prompts: Details of the instruction prompts used are described in Appendix A.
- Hardware: Experiments were conducted on a single NVIDIA H100 96GB GPU.

## 8 ETHICS STATEMENT

This work builds on publicly available datasets, including Natural Questions, BEIR, and MLDR, all of which are widely used in information retrieval research. We follow standard practices for data usage, and no personally identifiable information (PII) is involved. Our methods require access to internal attention scores of large language models, which may raise deployment constraints for proprietary systems; however, all experiments in this paper use open-weight models. As with any retrieval technology, potential risks include amplifying biases present in the underlying datasets or misranking relevant information in high-stakes domains. We encourage future work to investigate fairness, robustness, and bias mitigation in attention-based re-ranking methods before adoption in sensitive applications.

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

## A  PROMPT STRUCTURE

We use the same prompt structure for both head detection and re-ranking steps, which is adopted from Chen et al. (2025). The prompt starts with the instruction `Here are some paragraphs:` and the list of retrieved documents separated by a newline. After the list of document, there is a second prefix prompt for the query `Please find information that are relevant to the following query in the paragraphs above.  Query:`, and the query is appended at the end. The entire prompt string is wrapped by the special tokens `start token` and `end token` which vary based on the LLM used. See Table 5 for an example.

Table 5: An example prompt from NQ with Mistral 7B.

```
[INST] Here are some paragraphs:

[document 1] Can't Help Falling in Love "Can't Help Falling in Love" is a pop ballad
originally recorded by American singer Elvis Presley and published by Gladys Music,
Presley's publishing company. It was written by Hugo Peretti, Luigi Creatore, and
George David Weiss. The song was featured in Presley's 1961 film, Blue Hawaii.
During the following four decades, it was recorded by numerous other artists, including
Tom Smothers, Swedish pop group A-Teens, and the British reggae group UB40,
whose 1993 version topped the U.S. and UK charts.

[document 2] Can't Help Falling in Love In 2015, the song was included on the If I
Can Dream album, on the occasion of the 80th anniversary of Presley's birth. The version
uses archival voice recordings of Presley and his singers, backed by new orchestral
arrangements performed by the Royal Philharmonic Orchestra.

...

[document 40] I Can't Help It (If I'm Still in Love with You) Williams sang the song
with Anita Carter on the Kate Smith Evening Hour on April 23, 1952. The rare television
appearance is one of the few film clips of Williams in performance.

Please find information that are relevant to the following
query in the paragraphs above.

Query: who recorded i can't help falling in love with you [/INST]
```

## B  DISTRIBUTION OF RETRIEVAL HEADS

### B.1  RETRIEVAL HEAD LOCATION

Prior work Wu et al. (2025) show that only the top $10-15$ heads demonstrate strong retrieval signals. Our empirical results on the optimal number of heads (Section C.8) support this observation, as the re-ranking accuracy peaks within the top-10 heads and starts to decline after 10 heads. This indicates the top few heads are the most important heads for retrieval tasks, while heads outside of top-10 are less effective which may contribute noise to the retrieval process.

We list below the exact location of the top 8 NIAH retrieval heads, QR heads (without calibration) and CoRe heads for Llama-3.1 8B and Mistral 7B. We format $(L\text{-}H)$ as layer $L$ head $H$, for example, (7-19) means layer 7 head 19.

Llama-3.1 8B:

- NIAH retrieval heads: (15-30), (27-7), (8-1), (16-1), (24-27), (16-20), (5-8), (16-23).
- QR heads: (13-18), (14-13), (13-1), (20-14), (14-29), (16-1), (14-22), (17-29).
- CoRe heads: (13-18), (13-1), (14-13), (13-21), (14-31), (13-13), (8-11), (14-20).

Mistral 7B:

- NIAH retrieval heads: (18-0), (12-7), (12-6), (18-2), (18-3), (18-1), (30-8), (28-0).
- QR heads: (18-22), (15-26), (20-17), (18-0), (19-9), (16-22), (16-12), (19-16).
- CoRe heads: (15-21), (15-1), (16-12), (15-7), (9-26), (12-11), (12-7), (18-0).

We found that for Mistral 7B, CoRe heads overlap in 2 heads with both QR heads and NIAH retrieval heads. For Llama-3.1 8B, our top CoRe heads overlap with QR heads in 3 heads, but do not overlap with NIAH retrieval heads. Notably, the common 3 heads of CoRe and QR are in the top-3 and these are the heads with highest retrieval score. This explains why there is a close gap between QR-R and CoRe-R in the Llama results. Nevertheless, due to the difference in the remaining 5 heads, CoRe-R shows superior performance in many datasets that exhibit high number of hard negatives. In both models, CoRe heads appear in earlier layers compared to other heads, allowing for better accuracy-efficiency tradeoff with layer pruning.

| Baselines | Mistral Top-32 | Mistral Top-64 | Llama Top-32 | Llama Top-64 |
|---|---|---|---|---|
| NIAH heads | 19 | 35 | 16 | 30 |
| QR heads | 25 | 48 | 27 | 53 |

Table 6: Number of overlapping heads between CoRe heads and other baselines.

We also investigate the overlapping in top-32 and top-64 heads between CoRe heads and the baselines in Table 6. In both top-32 and top-64 heads, we found that CoRe heads overlaps in approximately 50% with NIAH and 75% with QR. However, as mentioned above, these lower-ranking heads are not important for retrieval tasks, and their retrieval scores are very low compared to the first top-8 heads.

## B.2 EFFECT OF CONTRASTIVE TEMPERATURE

As discussed in Section 4, the contrastive temperature $t$ controls the sharpness of the head distribution. Figure 9 demonstrates the effect of the temperature on the distribution of CoRe heads within Mistral 7B. Lower temperature heavily penalizes the non-CoRe heads which leads to larger gap in $S_{CoRe}$. Nonetheless, all levels of temperatures result in similar distribution of the top CoRe heads.

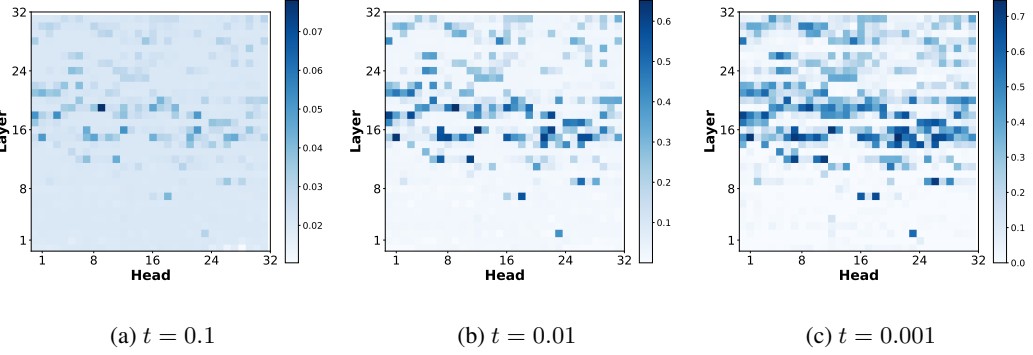

(a) $t = 0.1$        (b) $t = 0.01$        (c) $t = 0.001$

Figure 9: Distribution of $S_{CoRe}$ for all heads in Mistral 7B with different temperature $t$. All temperatures result in similar distribution of CoRe heads with the top few heads lie in the middle layers.

We also report the exact CoRe heads detected in Mistral 7B and Llama-3.1 8B with varying temperature below.

Llama-3.1 8B CoRe heads:

- $t = 0.1$: (13-18), (13-1), (14-13), (13-21), (14-31), (13-13), (8-11), (14-20).
- $t = 0.01$: (13-18), (14-13), (13-21), (14-31), (13-1), (14-20), (13-13), (13-3).
- $t = 0.001$: (13-21), (13-18), (14-31), (14-13), (14-20), (13-3), (13-1), (13-13).

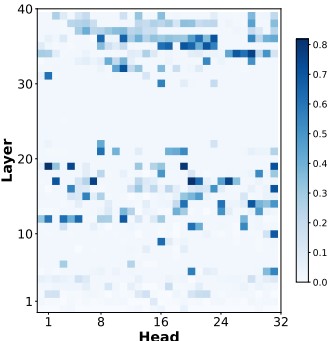

Figure 10: Distribution of $S_{CoRe}$ for all heads in Granite-3.2 8B with temperature $t = 0.001$. The top CoRe heads are in middle layers or late layers.

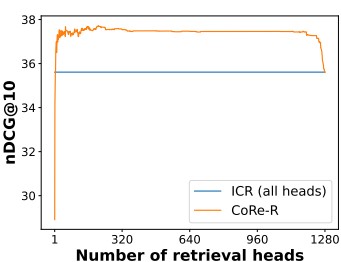

Figure 11: nDCG@10 on DBPedia top-40 with CoRe-R for Granite-3.2 8B. All attention heads within Granite-3.2 8B are equally important for re-ranking tasks.

Mistral 7B CoRe heads:

- $t = 0.1$: (19-9), (18-3), (16-12), (18-0), (16-22), (18-22), (15-1), (15-8).
- $t = 0.01$: (19-9), (16-12), (15-1), (16-22), (15-21), (12-11), (18-0), (18-22).
- $t = 0.001$: (15-21), (15-1), (16-12), (15-7), (9-26), (12-11), (12-7), (18-0).

## C  ADDITIONAL EXPERIMENT RESULTS

### C.1  RESULTS ON GRANITE-3.2 8B MODEL

In this subsection, we provide experimental results on Granite-3.2 8B model. Figure 10 demonstrates the distribution of CoRe heads within Granite-3.2 8B model, and the specific CoRe heads are:

$$(19\text{-}1), (17\text{-}20), (19\text{-}19), (34\text{-}28), (17\text{-}25), (17\text{-}7), (19\text{-}4), (19\text{-}31).$$

Beside the CoRe heads concentrated in the middle layers, many heads in the late layers also exhibit high $S_{CoRe}$ which can potentially be considered CoRe heads. Nevertheless, most of the top 8 CoRe heads still lie in the middle layers for Granite, with an exception for head (34-28).

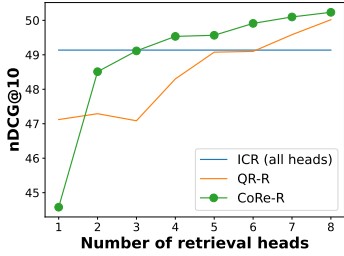

Figure 12: Average nDCG@10 on BEIR benchmark for Granite-3.2 8B using attention signal from different number of top retrieval heads.

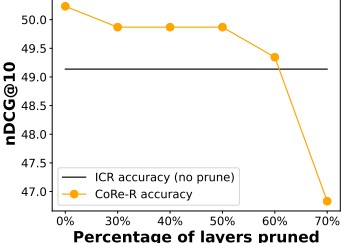

Figure 13: Average nDCG@10 on BEIR benchmark for Granite-3.2 8B with different levels of layer pruning.

Figure 11 shows the nDCG@10 score on DBPedia dataset for Granite-3.2 8B with different number of retrieval heads. Similar to other models, we observe that the attention signal from fewer number of heads achieves better re-ranking accuracy than the noisy aggregation over all heads (ICR), and the nDCG@10 score peaks with a small number of CoRe heads. Interestingly, there is negligible accuracy degradation across different number of retrieval heads. This trend indicates that most

attention heads within Granite-3.2 8B do not contribute much noise to the aggregated attention signals. Hence, the activation of more attention heads has a minimal influence on the final re-ranking accuracy. Figure 12 provides a closer look at the top 8 CoRe heads with comparison to ICR and QR-R. Again, CoRe-R with more than 3 heads consistently outperforms ICR and QR-R.

Figure 13 demonstrates the affect of layer pruning on the re-ranking accuracy of CoRe-R with Granite-3.2 8B. We observe that pruning $30 - 50\%$ of the final layers results in nearly identical nDCG@10 with a slight drop of $0.4$ points compared to no pruning. This result highlights the importance of the late CoRe head (34-28). Similar to other models, there is a decline in re-ranking performance with more than $50\%$ layers pruned, underscoring the importance of the CoRe heads in the middle layers.

## C.2 RESULTS WITH CUSTOMIZED PROMPTS

Table 7: nDCG@10 on Quora dataset with the universal prompt and the customized prompt.

| Method | Llama-3.1 8B | | Phi-4 | | Granite-3.2 8B | |
|---|---|---|---|---|---|---|
| | universal | customized | universal | customized | universal | customized |
| ICR | 78.7 | 83.8 | 74.2 | 76.8 | **83.1** | 83.8 |
| QR-R | 74.6 | 80.8 | 76.5 | 80.7 | 79.8 | 82.3 |
| CoRe-R | **82.3** | **85.4** | **80.2** | **85.4** | 75.1 | **84.4** |

All attention-based re-rankers underperform the retriever baseline in some datasets that require more task-specific instruction prompts. For example, we show that a customized prompt for duplicated question dataset Quora, i.e. `Please identify question that has the exact same meaning with the following query. Query:` can improve the performance.

We report the re-ranking results in Table 7. Overall, the task-specific prompt increases the nDCG@10 scores for all attention-based re-rankers by large margin. As expected, CoRe-R still stands out as the best-performing re-ranker, delivering the best re-ranking accuracy across all models. Future work could explore different ways to optimize the task-specific prompt design for Quora as well as other datasets, which further improve the final re-ranking performance.

## C.3 RESULTS ON STATISTICAL SIGNIFICANCE

We conducted a randomized stratified hypothesis test (Noreen, 1989), comparing CoRe-R and QR-R. For each query, we randomly permute their scores with 50% probability and compute the distribution of permuted differences. The p-value is the proportion of permutations in which the permuted difference exceeds the original difference. As shown in Table 8, 9, all LLMs evaluated on BEIR and MLDR yield p-values well below $0.05$, confirming that CoRe-R's improvements over QR-R are statistically significant.

Table 8: Statistical significance testing of CoRe-R vs. QR-R on the BEIR benchmark.

| | Mistral 7B | Llama-3.1 8B | Phi-4 | Granite-3.2 8B |
|---|---|---|---|---|
| Core-R − QR-R | 2.0 | 0.8 | 0.4 | 0.3 |
| p-value | 0.001 | 0.001 | 0.001 | 0.033 |

Table 9: Statistical significance testing of CoRe-R vs. QR-R on the MLDR benchmark.

| | Mistral 7B | Llama-3.1 8B | Granite-3.2 8B |
|---|---|---|---|
| Core-R − QR-R | 4.3 | 1.1 | 0.6 |
| p-value | 0.001 | 0.001 | 0.034 |

## C.4 RESULTS ON LAYER PRUNING

We report the layer pruning experiment for the baselines NIAH-R and QR-R in Figure 14. As reported in Section B.1, NIAH and QR contain many mid-layer heads, NIAH-R and QR-R also

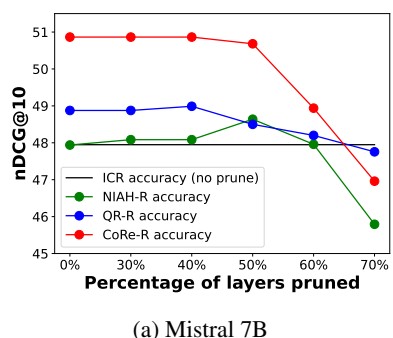

(a) Mistral 7B

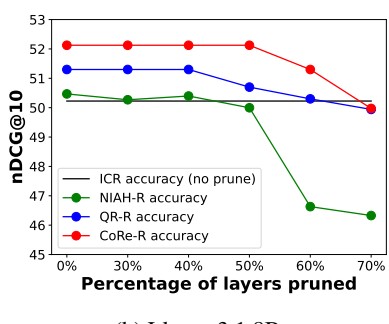

(b) Llama-3.1 8B

Figure 14: Average re-ranking accuracy on BEIR benchmark with different levels of layer pruning.

show efficiency benefit with 30-40% layer pruning. However, both baselines start to degrade much earlier (after 40%) compared to CoRe-R which continues to retain near-identical performance up to 50%. This observation makes intuitive sense because all CoRe heads lie in early to mid layers while NIAH and QR have some late-layer heads (layer 20 and after). Overall, CoRe-R not only delivers the highest re-ranking accuracy, but also shows the best accuracy-efficiency tradeoff thanks to its structural head location.

## C.5 RESULTS ON MULTI-HOP TASKS

We conduct additional experiment on two multi-hop tasks MuSiQue (Trivedi et al., 2022) and CLIP-PER (Pham et al., 2025). We follow the same setup in Chen et al. (2025) for MuSiQue dataset and adopt the same setting in Zhang et al. (2025) for CLIPPER dataset.

Table 10: Re-ranking performance on multi-hop datasets.

| Dataset | Retriever Baseline | Mistral 7B | | | Llama-3.1 8B | | | Phi-4 | | | Granite-3.2 8B | | |
|---|---|---|---|---|---|---|---|---|---|---|---|---|---|
| | | ICR | QR-R | CoRe-R | ICR | QR-R | CoRe-R | ICR | QR-R | CoRe-R | ICR | QR-R | CoRe-R |
| MuSiQue (R@2) | 37.9 | 40.0 | 41.9 | **43.9** | 44.5 | 45.9 | **46.7** | 40.9 | 45.9 | **46.3** | 45.7 | 46.1 | **46.8** |
| MuSiQue (R@5) | 49.2 | 53.9 | 54.4 | **56.2** | 57.0 | 58.3 | **58.9** | 54.8 | 57.3 | **58.5** | 56.4 | 57.2 | **58.4** |
| CLIPPER (R@2) | 5.1 | 24.2 | 25.9 | **26.5** | 27.8 | 29.0 | **29.8** | 26.5 | 28.3 | **28.9** | 26.6 | 26.7 | **27.5** |
| CLIPPER (R@5) | 22.0 | 43.0 | 45.1 | **45.7** | 45.7 | **48.8** | 48.6 | 46.5 | 47.2 | **47.9** | 46.1 | 44.6 | **47.1** |

Table 10 reports Recall@2 and Recall@5 for both multi-hop datasets with four language models. Overall, CoRe-R stands out with the best re-ranking accuracy in both tasks across all models with only one exception of CLIPPER Recall@5 in Llama-3.1 8B.

## C.6 RESULTS ON LONG CONTEXT

In this subsection, we evaluate the attention-based re-rankers on various context length with the BEIR benchmark. We report per-dataset nDCG@10 using top-60 documents in Table 11, top-80 documents in Table 12 and top-100 documents in Table 13.

In both context length settings, CoRe-R outperforms the baselines across all models. Similar to the re-ranking results for top-40 in the main text, CoRe-R shows prominent improvement in all datasets for Mistral 7B. For Llama-3.1 8B, CoRe-R even shows further improvement with longer context length on many datasets compared to top-40, while both baselines ICR and QR-R show accuracy degradation. These results widen the gap between CoRe-R and the baselines, demonstrating the superior robustness of CoRe-R in long context settings.

## C.7 RESULTS ON DIFFERENT DETECTION DATASETS

To study the sensitivity of CoRe-R to the detection dataset, we conduct the re-ranking experiments using different CoRe heads detected using a subset of the MSMARCO dataset. We found that the

Table 11: nDCG@10 on the BEIR benchmark top-60.

| Dataset | Retriever Baseline | Mistral 7B | | | Llama-3.1 8B | | | Phi-4 | | | Granite-3.2 8B | | |
|---|---|---|---|---|---|---|---|---|---|---|---|---|---|
| | | ICR | QR-R | CoRe-R | ICR | QR-R | CoRe-R | ICR | QR-R | CoRe-R | ICR | QR-R | CoRe-R |
| TREC-COVID | 63.1 | 71.5 | 74.3 | **74.6** | 76.9 | 77.5 | **77.9** | 73.8 | **77.0** | 77.0 | **73.7** | 71.7 | 71.9 |
| NFCorpus | 33.7 | 32.7 | 34.3 | **34.8** | 34.5 | 35.5 | **36.8** | 27.2 | 34.6 | **34.8** | 32.4 | **33.0** | 33.0 |
| DBPedia | 36.0 | 35.3 | 36.9 | **37.5** | 38.6 | 39.3 | **39.8** | 36.4 | 40.0 | **40.3** | 35.9 | 35.3 | **36.6** |
| SciFact | 71.3 | 72.1 | 71.3 | **73.6** | 74.3 | 74.5 | **75.3** | 65.2 | 70.3 | **70.9** | 75.1 | 74.2 | **74.5** |
| SciDocs | 22.5 | 16.7 | 17.1 | **19.3** | 18.5 | 18.6 | **19.6** | 17.5 | **20.7** | 20.7 | 19.4 | 19.5 | **19.6** |
| FiQA | 36.9 | 37.0 | 39.6 | **41.6** | 42.0 | 42.7 | **44.2** | 36.0 | 42.8 | **43.0** | 41.1 | 40.9 | **41.9** |
| NQ | 51.6 | 53.1 | 54.9 | **56.8** | 61.0 | **63.3** | 63.2 | 56.9 | 63.1 | **63.7** | 59.5 | 57.9 | **60.5** |
| FEVER | 85.5 | 87.4 | 87.0 | **88.4** | **89.1** | 88.2 | 88.1 | 89.0 | **89.6** | 89.6 | **88.4** | 88.0 | 86.6 |
| Climate-FEVER | 30.3 | 21.4 | 20.8 | **23.2** | 22.3 | 21.7 | **22.6** | 22.4 | **24.9** | 24.5 | **21.9** | 21.1 | 21.0 |
| HotpotQA | 62.9 | 72.2 | 72.5 | **74.2** | 74.9 | 74.6 | **75.0** | 74.4 | 75.6 | **75.7** | 72.0 | 73.3 | **73.6** |
| Touche | 24.0 | 23.5 | 25.8 | **27.6** | 26.9 | 27.4 | **28.2** | 19.2 | 25.3 | **25.6** | 23.5 | 25.2 | 25.6 |
| MSMARCO | 30.7 | 28.8 | **31.1** | 31.0 | 32.8 | 34.5 | **34.8** | 30.1 | 34.5 | **34.7** | 30.7 | 30.4 | **31.8** |
| Quora | 86.7 | 81.2 | 71.4 | **85.1** | 78.0 | 72.9 | **81.5** | 75.2 | 70.8 | **80.5** | **79.5** | 70.1 | 74.5 |
| ArguAna | 56.4 | 46.6 | 51.2 | **53.3** | 43.0 | 50.5 | **55.1** | 40.7 | 49.8 | **51.7** | 54.7 | 54.6 | **56.7** |
| CQADupstack | 44.3 | 38.3 | 40.3 | **41.2** | 41.7 | 43.1 | **43.4** | 37.3 | 42.9 | **43.5** | 42.3 | 43.0 | **43.6** |
| Average | 49.1 | 47.8 | **48.6** | 50.8 | 50.2 | 50.9 | **52.4** | 46.7 | 50.8 | **51.7** | 50.0 | 49.2 | **50.1** |

Table 12: nDCG@10 on the BEIR benchmark top-80.

| Dataset | Retriever Baseline | Mistral 7B | | | Llama-3.1 8B | | | Phi-4 | | | Granite-3.2 8B | | |
|---|---|---|---|---|---|---|---|---|---|---|---|---|---|
| | | ICR | QR-R | CoRe-R | ICR | QR-R | CoRe-R | ICR | QR-R | CoRe-R | ICR | QR-R | CoRe-R |
| TREC-COVID | 63.1 | 69.6 | 74.0 | **74.8** | **78.4** | 78.3 | 78.2 | 73.1 | **77.5** | 77.5 | 73.4 | 70.4 | **73.7** |
| NFCorpus | 33.7 | 32.7 | 33.7 | **33.9** | 33.4 | 35.3 | **36.8** | 23.8 | **32.9** | 32.6 | 31.6 | 31.8 | **32.6** |
| DBPedia | 36.0 | 34.6 | 36.8 | **37.6** | 39.0 | 39.5 | **40.0** | 35.8 | **40.5** | 40.5 | 36.7 | 35.5 | **36.4** |
| SciFact | 71.3 | 70.6 | 67.1 | **71.5** | 73.5 | 74.0 | **76.0** | 53.6 | 60.2 | **61.6** | 74.5 | 73.8 | 74.1 |
| SciDocs | 22.5 | 16.4 | 17.1 | **19.0** | 18.1 | 18.2 | **19.4** | 15.8 | 19.0 | **19.8** | 19.4 | **19.5** | 19.1 |
| FiQA | 36.9 | 36.1 | 40.1 | **41.7** | 41.3 | 42.3 | **43.5** | 33.3 | **42.0** | 41.8 | 40.4 | 39.8 | **41.1** |
| NQ | 51.6 | 51.3 | 54.5 | **56.3** | 60.2 | **63.5** | 62.8 | 56.1 | 63.2 | **63.3** | 59.3 | 57.7 | **60.3** |
| FEVER | 85.5 | 87.3 | 86.9 | **88.5** | 86.0 | 86.1 | **87.9** | 88.5 | 89.2 | **89.3** | 87.3 | 87.1 | 87.1 |
| Climate-FEVER | 30.3 | 21.1 | 20.9 | **22.7** | 22.2 | **22.5** | 22.4 | 21.5 | 24.0 | **24.3** | 22.2 | 21.2 | 21.3 |
| HotpotQA | 62.9 | 72.5 | 72.9 | **74.8** | 75.6 | 75.4 | **75.8** | 74.9 | **76.2** | 76.2 | 75.6 | 74.6 | 74.2 |
| Touche | 24.0 | 22.9 | 26.6 | **28.5** | 27.9 | 27.1 | **28.7** | 19.3 | 26.3 | **26.6** | 21.8 | 23.8 | **24.2** |
| MSMARCO | 30.7 | 27.9 | 28.7 | **30.4** | 32.7 | 34.4 | **34.7** | 29.7 | **34.6** | 34.6 | 30.7 | 30.2 | **32.2** |
| Quora | 86.7 | 81.3 | 71.7 | **85.1** | 79.0 | 72.6 | **82.4** | 74.6 | 74.4 | **78.6** | 73.5 | 72.6 | **74.9** |
| ArguAna | 56.4 | 48.2 | 52.3 | **53.1** | 43.2 | 50.7 | **55.2** | 36.6 | 47.8 | **48.2** | 54.3 | 55.1 | **55.7** |
| CQADupstack | 44.3 | 38.0 | 40.1 | **40.7** | 41.6 | 42.9 | **43.2** | 36.3 | 42.1 | **42.4** | 42.2 | 42.0 | **42.9** |
| Average | 49.1 | 47.4 | 48.2 | **50.6** | 50.1 | 50.8 | **52.5** | 44.9 | 49.9 | **50.5** | 49.5 | 49.0 | **50.0** |

Table 13: nDCG@10 on the BEIR benchmark top-100.

| Dataset | Retriever Baseline | Mistral 7B | | | Llama-3.1 8B | | | Phi-4 | | | Granite-3.2 8B | | |
|---|---|---|---|---|---|---|---|---|---|---|---|---|---|
| | | ICR | QR-R | CoRe-R | ICR | QR-R | CoRe-R | ICR | QR-R | CoRe-R | ICR | QR-R | CoRe-R |
| TREC-COVID | 63.1 | 66.8 | 73.0 | **73.7** | 76.2 | 79.2 | **79.6** | 71.0 | 79.0 | **79.2** | 72.9 | 71.6 | **75.0** |
| NFCorpus | 33.7 | 31.5 | 31.3 | **32.5** | 33.2 | 35.2 | **36.8** | 18.3 | 23.3 | **23.4** | 31.9 | 31.8 | **32.7** |
| DBPedia | 36.0 | 33.6 | 36.7 | **37.5** | 38.9 | 39.7 | **39.8** | 35.3 | **40.5** | 40.5 | 36.5 | 36.1 | **37.4** |
| SciFact | 71.3 | 62.9 | 58.2 | **66.0** | 74.1 | 74.8 | **76.7** | 41.3 | 45.4 | **45.9** | 74.0 | 74.0 | 73.8 |
| SciDocs | 22.5 | 16.5 | 16.5 | **18.6** | 17.5 | 18.2 | **19.3** | 10.1 | 15.1 | **15.2** | 19.4 | **19.5** | 19.3 |
| FiQA | 36.9 | 35.6 | 39.5 | **40.3** | 41.0 | 42.0 | **43.9** | 27.7 | 36.3 | **36.8** | 40.4 | 39.7 | **41.0** |
| NQ | 51.6 | 50.2 | 53.8 | **55.5** | 60.7 | 60.9 | **62.4** | 54.8 | 62.8 | **62.9** | 58.8 | 57.0 | **60.3** |
| FEVER | 85.5 | 87.1 | 86.4 | **88.2** | **88.9** | 88.0 | 87.8 | 87.2 | **88.8** | 88.6 | 86.5 | 87.1 | **87.4** |
| Climate-FEVER | 30.3 | 21.4 | 21.5 | **23.1** | 22.3 | 22.1 | **22.6** | 17.9 | 21.3 | **21.7** | **22.1** | 21.0 | 21.9 |
| HotpotQA | 62.9 | 72.7 | 73.2 | **75.1** | 76.0 | 75.8 | **76.1** | 75.4 | 76.7 | **76.8** | 73.4 | 74.1 | **75.2** |
| Touche | 24.0 | 21.6 | 26.4 | **27.1** | 26.9 | 26.1 | **27.3** | 14.7 | **24.6** | 24.3 | 21.2 | 23.1 | **23.7** |
| MSMARCO | 30.7 | 27.3 | 28.5 | **30.2** | 32.3 | **34.2** | 34.2 | 28.9 | 34.4 | **34.5** | 30.2 | 31.0 | **32.1** |
| Quora | 86.7 | 81.5 | 72.4 | **85.1** | 78.9 | 71.7 | **82.0** | 75.3 | 76.1 | **78.4** | 73.7 | 74.9 | **75.7** |
| ArguAna | 56.4 | 48.5 | **52.4** | 52.4 | 42.5 | 50.3 | **54.9** | 17.1 | 32.2 | **32.3** | 53.9 | 54.4 | **55.2** |
| CQADupstack | 44.3 | 37.6 | 39.2 | **39.8** | 41.3 | 42.5 | **43.0** | 32.9 | **40.9** | 39.6 | 41.9 | 42.0 | **42.7** |
| Average | 49.1 | 46.3 | 47.3 | **49.7** | 50.0 | 50.7 | **52.4** | 40.5 | 46.4 | **46.7** | 49.1 | 49.1 | **50.2** |

new detected CoRe heads are nearly identical to the CoRe heads detected using NQ, with 7 out of 8 overlapping heads. Specifically, the exact head location is listed below.

Llama-3.1 8B CoRe heads:

- Detected via NQ: (13-18), (13-1), (14-13), (13-21), (14-31), (13-13), (8-11), (14-20).

Table 14: nDCG@10 of CoRe-R on BEIR benchmark with different hard negatives data.

| Dataset | Retriever Baseline | Mistral 7B | | Llama-3.1 8B | |
|---|---|---|---|---|---|
| | | NQ | MSMARCO | NQ | MSMARCO |
| TREC-COVID | 63.1 | 73.8 | 74.5 | 75.7 | 75.6 |
| NFCorpus | 33.7 | 35.2 | 35.6 | 36.7 | 36.5 |
| DBPedia | 36.0 | 37.5 | 38.1 | 39.4 | 39.4 |
| SciFact | 71.3 | 74.7 | 73.6 | 75.1 | 75.8 |
| SciDocs | 22.5 | 19.5 | 19.0 | 20.2 | 20.0 |
| FiQA | 36.9 | 41.4 | 41.3 | 44.1 | 43.9 |
| NQ | 51.6 | 58.1 | 58.3 | 63.2 | 63.3 |
| FEVER | 85.5 | 88.7 | 88.3 | 88.4 | 88.7 |
| Climate-FEVER | 30.3 | 23.3 | 22.7 | 22.7 | 22.7 |
| HotpotQA | 62.9 | 73.4 | 73.0 | 73.8 | 73.9 |
| Touche | 24.0 | 26.2 | 27.4 | 26.4 | 26.3 |
| MSMARCO | 30.7 | 31.7 | 32.3 | 34.9 | 35.1 |
| Quora | 86.7 | 85.2 | 83.8 | 82.3 | 82.0 |
| ArguAna | 56.4 | 52.7 | 52.2 | 55.0 | 54.7 |
| CQADupstack | 44.3 | 41.6 | 41.4 | 43.9 | 43.9 |
| Average | 49.1 | 50.9 | 50.8 | 52.1 | 52.1 |

- Detected via MSMARCO: (13-18), (13-1), (14-13), (13-21), (14-31), (16-1), (8-11), (14-20).

Mistral 7B CoRe heads:

- Detected via NQ: (15-21), (15-1), (16-12), (15-7), (9-26), (12-11), (12-7), (18-0).
- Detected via MSMARCO: (15-21), (15-1), (16-12), (15-7), (9-26), (12-11), (19-9), (18-0).

We further repeat the re-ranking experiment on the BEIR benchmark using the CoRe heads detected via MSMARCO. The main results in Table 14 show similar and consistent performance, confirming that CoRe heads do not heavily depend on the detection data.

## C.8 RESULTS ON DIFFERENT NUMBER OF CORE HEADS

We repeat the "number of heads" experiment from Figure 2 on three additional datasets NQ, NF-Corpus and FiQA using two models Mistral 7B and Llama-3.1 8B. Figure 15 shows the re-ranking results in each setting, and we observe the same consistent trend where the accuracy peaks within the top-10 CoRe heads and gradually declines as the number of heads grows. This stability across diverse datasets supports our claim that CoRe requires no dataset-specific tuning and that a small, fixed number of heads suffices in practice.

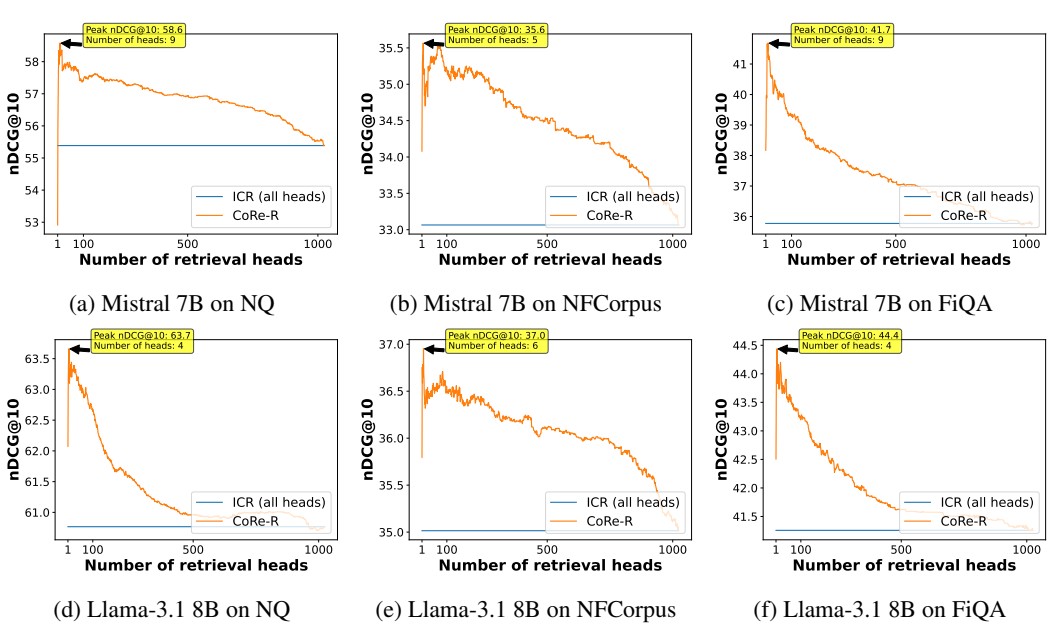

(a) Mistral 7B on NQ  (b) Mistral 7B on NFCorpus  (c) Mistral 7B on FiQA

(d) Llama-3.1 8B on NQ  (e) Llama-3.1 8B on NFCorpus  (f) Llama-3.1 8B on FiQA

Figure 15: nDCG@10 with CoRe-R across all number of CoRe heads.

