# OpenReview forum: "Less is More: Contrastive Retrieval Heads Improve Attention-Based Re-Ranking"
_ICLR.cc/2026/Conference — Submitted to ICLR 2026_

### Official Review · Reviewer_3Gtp · 2025-10-26

**Soundness:** 3
**Presentation:** 2
**Contribution:** 2
**Rating:** 4
**Confidence:** 4

**Summary:**

The paper studies the problem of building effective attention-based re-ranking systems. The paper proposes CoRe (Contrastive Retrieval Heads), which extends previous attention-based re-rankers by 1) using hard negative data for identifying important heads. introducing a contrastive head selection criterion and 2) CoRe explicitly re-normalizing attention to relevant versus irrelevant documents. The paper evaluates the method on the BEIR benchmark and the multilingual MLDR dataset across several open-weight models, showing some improvements over ICR and QRHead baselines.

**Strengths:**

The paper also considers  a simple yet effective layer-pruning strategy, showing that re-ranking accuracy can be preserved while cutting memory usage by ~40% and latency by ~20%.

The proposed extension is straightforward and seems to be effective.

Evaluation covers two re-ranking datasets (BEIR, MLDR) and four open-source LLMs.

The paper is mostly well presented as easy to understand.

**Weaknesses:**

I have some concerns regarding the framing of the core idea of CoRE. The paper suggests that QRHead’s absolute scoring fails to capture relative ranking among documents (Ln 146). As attention scores already result from a softmax normalization, they implicitly encode relative weighting. It seems the CoRe is renormalizing at the document level to ignore other parts like instructions, rather than introducing new contrastiveness.

The current evaluation focuses primarily on BEIR-style single-hop retrieval. Past work (e.g., ICR, QRHead) also explored multi-hop reasoning tasks such as MuSiQue and CLIPPER.

The paper could include more discussion regarding context length. One main advantage of attention-based re-rankers over generative methods is the potential to scale to long context settings. The experiment mainly considers reranking 40 documents. It would be good to include or discuss performance and scaling behavior under a larger number of documents.

The paper needs more analysis on intrinsic differences of CoRE heads and QRHeads. Like reporting overlapping between top-32, 64 heads.

**Questions:**

How does CoRe generalize to multi-hop reasoning?

How does CoRE generalize to longer contexts?

What’s the overlap between CoRE heads and QRHeads among top-32, 64 heads?

---

> ### Author Response · Authors · 2025-11-21
>
> ### 1. I have some concerns regarding the framing of the core idea of CoRE. The paper suggests that QRHead’s absolute scoring fails to capture relative ranking among documents (Ln 146). As attention scores already result from a softmax normalization, they implicitly encode relative weighting. It seems the CoRe is renormalizing at the document level to ignore other parts like instructions, rather than introducing new contrastiveness.
>
> We thank the reviewer for raising this point, and we would like to point our response to the **clarification on the methodology** in the general response. We want to clarify that CoRe does not ignore the instruction in the token-level attention computation. Like ICR and QR, the instruction attention score is only removed when computing the document-level scores. We agree that the token-level attention scores are produced by a softmax and therefore encode a form of relative weighting across all tokens in the input. However, this normalization operates at the *token* level, not at the *document* level, and the aggregated document-level scores may not preserve the normalization.
>
> QR-Head evaluates a head solely using the absolute document-level score of the  positive document and it does not compare positives and negatives. In contrast, CoRe’s contrastive scoring explicitly evaluates *relative* attention between the positive and hard-negative passages using an InfoNCE-style objective. This penalizes precisely those ambiguous heads that QR fails to distinguish.
>
> Thus, CoRe’s contribution is not a renormalization of the attention softmax, but a new retrieval-specific scoring criterion applied at the *document-level*, where absolute vs. relative discrimination directly affects ranking. We have clarified this distinction in the revised manuscript.
>
> ### 2. The current evaluation focuses primarily on BEIR-style single-hop retrieval. Past work (e.g., ICR, QRHead) also explored multi-hop reasoning tasks such as MuSiQue and CLIPPER.
>
> We thank the reviewer for the suggestion. In the revision, we added the experiments on both MuSiQue and CLIPPER datasets with three models Mistral, Llama and Phi. Due to limited space, we reported the results on MuSiQue in Section 5.2 (Figure 3) and included the results on CLIPPER in Appendix C.5. In almost all cases, CoRe-R shows the highest re-ranking accuracy, highlighting the applicability of CoRe heads to both single-hop and multi-hop tasks.
>
> ### 3. The paper could include more discussion regarding context length. One main advantage of attention-based re-rankers over generative methods is the potential to scale to long context settings. The experiment mainly considers reranking 40 documents. It would be good to include or discuss performance and scaling behavior under a larger number of documents.
>
> We thank the reviewer for this very helpful suggestion. We have added new re-ranking experiments on top-60, top-80 and top-100 documents on the BEIR benchmark. We included the average score for Mistral and Llama in Section 5.2 (Figure 4) and reported per-dataset nDCG@10 of all models in Appendix C.6. In longer context settings, we observe a decline in Mistral across all three methods, which aligns with prior observations that the Mistral 7B model is not robust under long-context
> settings (Scale AI, 2024). Nevertheless, CoRe-R still outperforms both baselines for Mistral in all context length settings. For Llama, similar drop in accuracy also occurs in ICR and QR, however, CoRe-R shows a slight but impressive increase in nDCG@10, widening the gap between CoRe-R and other baselines. We have added this discussion in Section 5.2 of the revision.
>
> *Scale AI. A guide to improving long context instruction following on open source models. https://scale.com/blog/long-context-instruction-following, 2024.*

---

> ### Author Response · Authors · 2025-11-21
>
> ### 4. The paper needs more analysis on intrinsic differences of CoRE heads and QRHeads. Like reporting overlapping between top-32, 64 heads.
>
> We thank the reviewer for the suggestion. Prior work (e.g., Wu et al. 2025) shows that only the top 10–15 heads exhibit meaningful retrieval behavior. Our empirical results confirm this: re-ranking accuracy consistently peaks within the top-10 heads (Figure 2 and Figure 15), and the nDCG@10 for CoRe-R decreases rapidly once more than the top-10 CoRe heads are included.
>
> Our original analysis of the top-8 heads (Section 5.4, Appendix B.1) already indicates minimal overlap across methods: CoRe shares at most 2–3 heads with QR and at most 0–1 head with NIAH. This demonstrates that the high-importance CoRe heads are intrinsically different from those selected by QR and NIAH.
>
> Following the reviewer’s request, we have added overlap statistics for the top-32 and top-64 heads (Appendix B.1). At these larger ranks, CoRe overlaps with NIAH by approximately 50\% and with QR by approximately 75\%. However, heads beyond the top-10 have very low and nearly indistinguishable retrieval scores, and thus contribute negligibly to performance. Consequently, the larger overlap at ranks 32 and 64 is largely attributable to randomness and does not reflect meaningful similarity among methods.
>
> Overall, the expanded analysis supports that the substantive differences occur among the top-ranked heads, which dominate retrieval quality, while overlap among lower-ranked heads carries limited interpretive value.
>
> *Wenhao Wu, et al. Retrieval head mechanis-
> tically explains long-context factuality. In The Thirteenth International Conference on Learning Representations, 2025.*
>
> ### 5. How does CoRe generalize to multi-hop reasoning?
>
> In addition to the multi-hop dataset HotpotQA in BEIR, we ran experiments on two additional multi-hop datasets MuSiQue and CLIPPER. We reported the results on MuSiQue in Figure 3 and CLIPPER in Appendix C.3. With the new multi-hop tasks, CoRe-R still delivers the best re-ranking performance compared to the strong baselines, indicating the applicability of CoRe heads in both single-hop and multi-hop retrieval tasks.
>
> ### 6. How does CoRE generalize to longer contexts?
>
> As mentioned in our response to point 3, CoRe-R consistently outperforms other baselines for Mistral in all context length settings. Impressively, CoRe-R even improves upon longer context accuracy in Llama when other baselines show degradation.
>
> ### 7. What’s the overlap between CoRE heads and QRHeads among top-32, 64 heads?
>
> As mentioned in our response to Point 4, there is about 50\% overlap between CoRe and NIAH, and 75\% overlap between CoRe and QR. However, we observe that the retrieval scores of the top-8 heads dominate others, and the remaining lower-ranking heads show very low and similar retrieval scores. Therefore, the ranking of the top-8 largely affects the retrieval ability while the ranking of heads outside of top-8 (i.e. top-32 and top-64) does not contain meaningful information about the retrieval ability of those heads.

---

### Official Review · Reviewer_ivKj · 2025-10-31

**Soundness:** 3
**Presentation:** 2
**Contribution:** 2
**Rating:** 4
**Confidence:** 4

**Summary:**

This paper introduces CoRe heads (Contrastive Retrieval heads), which is a subset of attention heads for document re-ranking. The key difference from previous work, such as QRHead, is the contrastive objective to find the retrieval heads. The CoRe heads are then used for document re-ranking, where the authors show improvement over existing approaches on the BEIR benchmark. The experiments include models from Llama, Phi, Mistral, and Granite model families. CoRe also uses fewer heads than other methods to achieve comparable or better performance.

**Strengths:**

The paper makes adjustments to existing approaches to detect existing approaches and achieve better performance on document re-ranking tasks. The experiments also covers multi-lingual settings that were not previously studied. The analysis also covers important system considerations such as the percentage of layers that can be pruned without affecting performance.

**Weaknesses:**

The contrastive scoring is mathematically similar to the objectives used in previous work. Specifically, the main baseline QR-Head uses post-softmax attention scores over the positive document. As a result, it’s achieves the same mathematical form as the paper’s retrieval score in equation 5 because the QRScore is normalized with respect to the attention mass assigned to the rest of the context as well. The only difference is that (1) CoRe scoring excludes the instruction from the overall attention mass, which is not carefully studied (would QR-Head perform the same if the instruction mass were excluded) and (2) CoRe does an additional softmax and tunes the temperature hyper parameters (the temperature in the QR-Head should also be tuned in a similar fashion to achieve fair comparison).
The paper could benefit from such theoretical analysis on how the mathematical form differ from previous works and where the improvements stem from.

Furthermore, while the results and analysis are strong, I find the novelty lacking as it only makes minor changes from previous works.

**Questions:**

How do the results scale with more heads beyond 8 heads?

How do the final results different with different temperature hyper parameters?

Why do CoRe-R work particularly well for Mistral but the results on the other models are much closer with QR-R?

---

> ### Author Response · Authors · 2025-11-21
>
> ### 1. The contrastive scoring is mathematically similar to the objectives used in previous work. Specifically, the main baseline QR-Head uses post-softmax attention scores over the positive document. As a result, it’s achieves the same mathematical form as the paper’s retrieval score in equation 5 because the QRScore is normalized with respect to the attention mass assigned to the rest of the context as well. The only difference is that (1) CoRe scoring excludes the instruction from the overall attention mass, which is not carefully studied (would QR-Head perform the same if the instruction mass were excluded) and (2) CoRe does an additional softmax and tunes the temperature hyper parameters (the temperature in the QR-Head should also be tuned in a similar fashion to achieve fair comparison). The paper could benefit from such theoretical analysis on how the mathematical form differ from previous works and where the improvements stem from.
>
> We thank the reviewer for raising this important point. A detailed clarification of the relationship between CoRe scoring, QR-Head, and the underlying attention normalization is provided in our general response under *Clarification on the methodology*. Briefly, CoRe computes document-level scores using the same attention signals as ICR and QR, and the exclusion of instruction tokens occurs only during the document-level aggregation step, not during the softmax computation itself. CoRe’s contrastive scoring further operates at the document-level and is therefore mathematically distinct from the token-level softmax used by QR-Head. Our revision includes an explicit mathematical comparison to avoid ambiguity and to highlight where the improvements stem from.
>
> ### 2. Furthermore, while the results and analysis are strong, I find the novelty lacking as it only makes minor changes from previous works.
>
> We thank the reviewer for raising this concern regarding novelty. While CoRe builds on the general idea of using attention heads for re-ranking, it introduces two substantive and novel contributions that go beyond prior work.
>
> First, our contrastive head-scoring mechanism is fundamentally different from both ICR and QR. Existing methods evaluate heads using only absolute attention to the positive document, which we show can misidentify heads that also attend strongly to hard negatives. Our work is the first to formulate head selection as a *relative, contrastive* problem using positive–negative discrimination. This conceptual shift leads to a new class of “contrastive retrieval heads” that are far more robust across datasets, models, and languages.
>
> Second, we are the first to investigate *layer pruning* in attention-based re-rankers. Prior work on retrieval heads (QR, NIAH) has not explored pruning. Our analysis reveals a new and previously unreported phenomenon: high-quality retrieval heads concentrate in earlier/middle layers, enabling up to 50\% pruning with minimal loss in accuracy, yielding a superior accuracy-efficiency tradeoff.
>
> Together, these contributions go beyond minor modifications. We have clarified these points in the revision to better highlight the novelty and impact of our approach.

---

> ### Author Response · Authors · 2025-11-21
>
> ### 3. How do the results scale with more heads beyond 8 heads?
>
> Thank you for the question. In the revision, we expanded the analysis beyond the DBPedia example in Figure 2. Specifically, we repeated the "number of heads'' sweep on three additional datasets (NQ, NFCorpus, and FiQA) using both Mistral 7B and Llama-3.1 8B. The new results (included in Appendix C.8) show the same consistent pattern: re-ranking accuracy increases as we add the top CoRe heads, peaks within approximately 8-10 heads, and gradually declines as more heads are included.
>
> This behavior aligns with our central hypothesis: while a small number of contrastive heads provide strong discriminative signals, adding lower-quality heads reintroduces the noise and redundancy that affect ICR, thereby reducing accuracy. The consistency of this trend across multiple datasets and two models suggests that the effectiveness of CoRe heads is concentrated in a very small subset, and that using fewer than 10 heads is a robust operating point.
>
> ### 4. How do the final results different with different temperature hyper parameters?
>
> To study the role of the temperature on the final re-ranking task, we performed ablation study on Mistral and Llama models with different temperatures. We reported the difference in detected heads and the re-ranking results in Section 5.5. We observe that many heads in Mistral attend strongly to both the positive document and hard negatives. Thus, higher temperatures ($t=0.1, 0.01$) were not able to capture high-quality heads, leading to different head sets compared to $t=0.001$. As a consequence, higher temperatures leads to lower re-ranking results for Mistral. Meanwhile, retrieval heads in Llama are more sparse, so Llama is less sensitive to the temperature compared to Mistral, and the final re-ranking accuracy is more stable across different temperatures. While temperature tuning is necessary for some LLMs, we note that the head detection is a one-time light-weight process which takes less than one hour on an H100 (96GB) GPU for the  models studied here.
>
> ### 5. Why do CoRe-R work particularly well for Mistral but the results on the other models are much closer with QR-R?
>
> We appreciate the reviewer’s question. The larger gains on Mistral arise from the model’s attention-head distribution. As mentioned in our response to point 4 above, it is harder to capture high-quality heads in Mistral as there are many "hard heads" that attend strongly to both positive and hard negatives. QR’s absolute scoring tends to select such ambiguous heads, while CoRe’s contrastive scoring explicitly penalizes them, leading to a larger improvement.
>
> For Llama and Phi which do not contain many hard heads, the QR-selected heads are already more selective and attend less to hard negatives, so QR-R starts from a stronger baseline and the margin for improvement is naturally smaller. Nevertheless, CoRe-R still achieves the best overall accuracy, and the relative gains vary with the model’s head patterns.

---

### Official Review · Reviewer_fEwK · 2025-11-01

**Soundness:** 3
**Presentation:** 3
**Contribution:** 3
**Rating:** 6
**Confidence:** 5

**Summary:**

In this paper, the authors propose to leverage a small subset of attention heads in LLMs to specialize in discriminating relevant from irrelevant documents for re-ranking tasks. Building on prior attention-based re-rankers and query-focused retrieval heads, they introduce a contrastive scoring metric inspired by InfoNCE loss. The experiments are conducted on the BEIR and MLDR benchmarks when the experimental results show improvements against the ICR baseline. Analysis also shows the approach can enable pruning of 50% of final layers with negligible accuracy loss, reducing inference latency by 20% and GPU memory by 40%, for the middle layers.

**Strengths:**

* Consistent improvements against the baseline method over diverse benchmarks
* By using only a tiny fraction of heads and enabling middle-layer focus for pruning, it significantly reduces latency, memory usage, and overhead compared to full-model aggregation, making it practical for real-world systems.
* The proposed approach reveals that discriminative retrieval signals are concentrated in middle layers, advancing interpretability of LLM attention and opening avenues for optimized architectures.

**Weaknesses:**

* The proposed approach highly relies on hard negatives from NQ, so the data quality could be sensitive.
* Hyperparameter tuning is required, and it adds the overhead and reduce the ease of use across new LLMs.
* The proposed method fails to beat baselines on certain datasets.

**Questions:**

* Could the optimal number of CoRe heads be different models and dataset? How sensitive is it?
* What specific criteria or thresholds are used to discard passages as false negatives during hard negative data construction?
* Can CoRe heads be dynamically updated or adapted for new datasets without re-running the entire detection process?

---

> ### Author Response · Authors · 2025-11-21
>
> ### 1. The proposed approach highly relies on hard negatives from NQ, so the data quality could be sensitive.
>
> We appreciate the reviewer’s concern. Although CoRe-R uses hard negatives from a small subset of the NQ training data for head detection, we found that the approach is **not** sensitive to this particular choice. The hard negatives serve only to guide which attention heads are most discriminative, and once the heads are selected, the re-ranking method is fully zero-shot and applied uniformly across all datasets. In addition, our contrastive scoring metric averages over thousands of head–sample interactions, which naturally smooths out occasional noisy or imperfect negatives.
>
> We also validated the robustness of CoRe by selecting a different detection dataset MSMARCO, and we found that the new detected CoRe heads are nearly identical to the CoRe heads detected using NQ, with 7 out of 8 overlapping heads for two models Mistral and Llama. We reported the re-ranking results using the new CoRe heads in Appendix C.7 and added a discussion in Section 5.5. Both sets of CoRe heads detected by NQ and MSMARCO show similar and consistent performance, confirming that CoRe heads do not heavily depend on the detection data.
>
> ### 2. Hyperparameter tuning is required, and it adds the overhead and reduce the ease of use across new LLMs.
>
> While CoRe does not rely on the detection data quality, we agree that the temperature needs to be tuned for new language models. However, we note that the head detection process is a **one-time** run per model as the detected CoRe heads show excellent generalizability to different retrieval tasks, including single-hop, multi-hop, cross-lingual and long context settings. In addition, the head detection process is relatively fast as it only requires a few thousand inferences with no training required. Throughout our extensive experiments on four large language models, we found that it takes less than an hour (H100 96GB GPU) to detect high-quality CoRe heads for each model. We have added this discussion to our ablation study on temperature in the revision (Section 5.5)
>
> ### 3. The proposed method fails to beat baselines on certain datasets.
>
> We acknowledge the reviewer’s observation that CoRe-R does not outperform the baselines on every individual dataset, and this behavior is expected in saturated benchmarks such as BEIR and MLDR where small fluctuations across subsets are common. Rather than winning every dataset, the key question is whether a method delivers **consistent and statistically reliable** improvements across models and tasks.
>
> To this end, we conducted a randomized stratified hypothesis test (Appendix C.3) comparing CoRe-R and QR-R. For each query, we randomly permute their scores with 50\% probability and compute the distribution of permuted differences. The p-value is the proportion of permutations in which the permuted difference exceeds the original difference. As shown in Appendix C.3, all four LLMs evaluated on BEIR and three LLMs evaluated on MLDR yield p-values well below 0.05, confirming that CoRe-R’s improvements over QR-R are **statistically significant** and unlikely to arise from noise.
>
> Beyond statistical significance, CoRe-R provides stable improvements across 15 BEIR datasets $\times$ 4 LLMs and 6 MLDR datasets $\times$ 3 LLMs, whereas QR-R frequently reduces accuracy and exhibits high variance. Furthermore, as shown in Figure 4 and Appendix C.6, when the retrieval depth increases from top-40 to top-60, top-80 and top-100, the performance gap between CoRe-R and both baselines **widens**. Larger candidate sets introduce more semantically similar hard negatives, under which QR-R’s absolute scoring becomes less selective and ICR’s raw attention becomes diluted. CoRe’s contrastive scoring maintains discriminability under these more challenging long-context conditions.
>
> Finally, CoRe-R achieves these gains while using fewer than 1\% of attention heads and supporting up to 50\% layer pruning, yielding a substantially more favorable accuracy--efficiency tradeoff.
>
> In summary, although CoRe does not obtain the best score on every dataset, it offers the most consistent, statistically validated, and long-context-robust improvements among the evaluated attention-based re-rankers.

---

> ### Author Response · Authors · 2025-11-21
>
> ### 4. Could the optimal number of CoRe heads be different models and dataset? How sensitive is it?
>
> We conducted additional experiments on three datasets (NQ, NFCorpus, FiQA) using two models (Mistral 7B and Llama-3.1 8B) to evaluate how the optimal number of CoRe heads varies across settings (Appendix C.8). Replicating the “number of heads’’ experiment from Figure 2, we observe a consistent pattern across all datasets and both models: re-ranking performance peaks within the first 10 CoRe heads. This matches our earlier finding on DBPedia and indicates that the effective number of heads is remarkably stable. These results suggest that CoRe does not require dataset-specific tuning and that a small, fixed number of heads is sufficient in practice.
>
> ### 5. What specific criteria or thresholds are used to discard passages as false negatives during hard negative data construction?
>
> We agree that the quality of hard negatives is crucial for reliable head detection. Our procedure is described in Section 4 (lines 212–215) in the revised paper. We summarize it here for clarity. For each of the 1000 samples from the Natural Questions (NQ) training set, we compute the retrieval score $S_{\mathrm{CoRe}}$ using all attention heads. Each sample consists of a query, a gold passage, and 49 hard negatives mined by the *ibm-granite/granite-embedding-30m-english* retriever. Hard negatives are drawn from the top 100 retrieved candidates.
>
> To further reduce false negatives, we follow deSouza P.Moreira et al. (2025) and discard any passage whose embedding similarity to the query is higher than that of the gold passage. This ensures that we do not incorrectly treat a potentially relevant document as a negative. The resulting dataset yields challenging but reliable hard negatives, enabling $S_{\mathrm{CoRe}}$ to be computed under conditions that favor the detection of genuinely discriminative and robust attention heads.
>
> *Gabriel de Souza P et al. Nv-retriever: Improving text embedding models with effective hard-negative
> mining, 2025*
>
> ### 6. Can CoRe heads be dynamically updated or adapted for new datasets without re-running the entire detection process?
>
> CoRe heads are designed to be detected once and then reused across tasks without any retraining. A key finding of our work is that the head sets identified from only 1,000 NQ examples generalize robustly to all 15 BEIR datasets and six MLDR languages, so re-running the detection process has not been necessary in practice.
>
> That said, if a user wishes to specialize CoRe heads for a new domain, the head-detection step can be re-run on a small set of domain-specific hard negatives, and this process is lightweight: it does not involve gradient updates, model fine-tuning, or changing model weights. We observe that the entire detection process takes less than an hour for all language models tested in our experiments. We have added this discussion in the revision (lines 227-229).

---

### Official Review · Reviewer_ipmp · 2025-11-01

**Soundness:** 3
**Presentation:** 3
**Contribution:** 2
**Rating:** 4
**Confidence:** 4

**Summary:**

The paper proposes a new attention-based re-ranker method that utilizes the attention weights of LLMs to perform re-ranking of documents. Built upon prior work on attention-based re-ranker methods, the paper introduces a novel approach to detect attention heads in LLMs that are important for re-ranking, termed as contrastive retrieval heads, by computing contrastive retrieval scores over positive and negative documents in the input: such heads attend to the irrelevant documents and are not distracted by the irrelevant documents. Empirical results show that re-ranking with contrastive heads outperforms other attention-based re-ranking baselines. Furthermore, such a method can be combined with layer pruning to improve the run-time efficiency as the detected contrastive heads concentrate in the middle layers of transformers.

**Strengths:**

1. The paper proposes a new re-ranking method that requires no training and demonstrates empirical results across multiple benchmarks with multiple LLMs.

2. The paper compares with several attention-based re-ranking baselines and shows strength over such baselines using the proposed method to detect important attention heads for re-ranking.

3. The paper further shows the strength of the method in terms of efficiency compared with baselines.

**Weaknesses:**

1. The performance of the contrastive retrieval head re-ranker is not particularly strong compared with the attention-based re-ranker baselines. In Table 1, the improvements of CoRe-R over ICR and QR-R are within around 1% for most subsets of the BEIR benchmark, and only for a single subset, Quora, can a relatively large improvement be observed. Similar results can be observed in Table 2 for the MLDR benchmark as well.

2. Related to question 1, there lacks an in-depth analysis of why CoRe-R outperforms the baselines only on certain tasks. Are there any insights or hypotheses on why CoRe-R performs particularly better than the baselines on the Quora subset of BEIR? What are some implications of such findings?

3. There lacks analysis of the similarities and differences between the heads detected by CoRe and by previous methods, especially NIAH and QR. Given the similar performance as mentioned in question 1, do they detect very similar attention heads? Does CoRe detect some very important missing heads that the other methods fail to detect?

4. There are missing baselines for the layer pruning experiments. Since the retrieval heads (i.e., NIAH-R) in Wu et al., 2024 also concentrate in the middle layers, I do not clearly see how layer pruning can only be applied to CoRe-R to improve efficiency, but not other attention-based re-rankers that use a sparse set of attention heads. Thus, the following baselines should be added for fairer evaluation of the efficiency benefit of the proposed method: QR-R (prune) and NIAH-R (prune).

**Questions:**

1. Why is contextual calibration only used for the re-ranking process, but not the head detection process of CoRe?

2. Figure 4: The head distribution in Mistral 7B is very different from those in the other models. Do you have any insights into this?

---

> ### Author Response · Authors · 2025-11-21
>
> ### 1. The performance of the contrastive retrieval head re-ranker is not particularly strong compared with the attention-based re-ranker baselines. In Table 1, the improvements of CoRe-R over ICR and QR-R are within around $1\%$ for most subsets of the BEIR benchmark, and only for a single subset, Quora, can a relatively large improvement be observed. Similar results can be observed in Table 2 for the MLDR benchmark as well.
>
> We thank the reviewer for the comment. While the absolute improvements of CoRe-R over ICR and QR-R on several BEIR subsets are within 0.5-1 nDCG@10, we first note that these gains are *statistically significant*. Using randomized stratified hypothesis testing, we find that for all four LLMs evaluated on BEIR and three LLMs evaluated on MLDR, the p-values for CoRe-R vs. QR-R are below 0.05 (see Appendix C.3), confirming that the improvements are unlikely to arise from random variation. This result directly addresses the concern regarding the magnitude of the difference.
>
> Beyond statistical significance, the strength of CoRe-R lies in its *consistency* across models and datasets. Across 15 BEIR datasets $\times$ 4 LLMs and 6 MLDR datasets $\times$ 3 LLMs, CoRe-R reliably outperforms ICR and QR-R, whereas QR-R frequently reduces accuracy and exhibits high variance.
>
> Importantly, CoRe-R’s advantage becomes *more pronounced* as the retrieval pool grows. As shown in Figure 4 and Appendix C.6, when increasing the number of retrieved documents from top-40 to top-60, top-80 and top-100, the performance gap between CoRe-R and both baselines consistently *widens* across nearly all datasets. This behavior aligns with the underlying difficulty of long-context retrieval: larger candidate sets introduce more semantically similar hard negatives, under which QR-R’s absolute scoring becomes less selective and ICR’s raw attention becomes diluted. CoRe’s contrastive scoring, by explicitly comparing positives against hard negatives, maintains discriminability under these more challenging settings.
>
> CoRe-R also achieves a superior efficiency-accuracy tradeoff. Because its heads predominantly lie in early/middle layers, the method supports up to 50\% layer pruning with negligible performance loss, yielding significant reductions in memory and latency. In contrast, ICR and QR-R rely on late-layer heads and degrade rapidly under pruning.
>
> Taken together, although the absolute gains are modest on some individual subsets, CoRe-R provides the most consistent, statistically reliable, and compute-efficient improvements among the evaluated attention-based re-rankers, with the advantage further increasing in long-context retrieval.
>
>
> ### 2. Related to question 1, there lacks an in-depth analysis of why CoRe-R outperforms the baselines only on certain tasks. Are there any insights or hypotheses on why CoRe-R performs particularly better than the baselines on the Quora subset of BEIR? What are some implications of such findings?
>
> The key idea behind CoRe-R is that its head-selection criterion explicitly distinguishes the positive document from hard negatives, i.e., irrelevant documents that are highly similar to the correct answer. Unlike ICR and QR-R, which rely on absolute attention mass and therefore do not penalize attention to confusing negatives, CoRe-R’s contrastive metric is designed precisely to surface heads that favor the positive while suppressing hard negatives.
>
> The Quora dataset is a duplicate-question retrieval task and is known to contain a high proportion of such hard negatives compared to other datasets. In Quora, many candidate questions share very similar lexical and semantic structure but differ subtly in meaning. This setting matches the strengths of CoRe’s contrastive scoring, which explains why the gains over ICR and QR-R are particularly pronounced on Quora.
>
> This observation provides a useful insight: the more a retrieval task is dominated by hard negatives, the more CoRe-R should outperform non-contrastive baselines. This suggests that contrastive head selection is especially valuable for domains with near-duplicate or semantically dense content (e.g., FAQ retrieval, paraphrase search, code retrieval), and highlights an advantage of CoRe-R that is not captured by average nDCG alone. We thank the reviewer for raising this insightful point. We have incorporated this explanation and the corresponding discussion into the revision.

---

> ### Author Response · Authors · 2025-11-21
>
> ### 3. There lacks analysis of the similarities and differences between the heads detected by CoRe and by previous methods, especially NIAH and QR. Given the similar performance as mentioned in question 1, do they detect very similar attention heads? Does CoRe detect some very important missing heads that the other methods fail to detect?
>
> We would like to clarify that the original manuscript already provided an analysis of head overlap across methods in Section 5.4, with the exact head indices (up to 8 heads) listed in Appendix B.1. For Llama-3.1-8B, we observe that CoRe and NIAH share no common heads, while CoRe and QR overlap in only 3 out of 8 heads. These 3 shared heads also rank among the top-3 for both methods, which helps explain the relatively small performance gap between CoRe-R and QR-R on Llama. Crucially, however, the remaining 5 CoRe heads are entirely distinct from QR’s selections. These CoRe-specific heads play an essential role in distinguishing positive from hard-negative passages, leading to substantially better performance on datasets such as Quora and in long-context settings (top-60, top-80 and top-100).
>
> We also highlight a structural difference: both NIAH and QR tend to select heads from later layers (layer 20 and above), whereas all CoRe heads lie in earlier layers (layer 18 and below). This difference is important for efficiency—because CoRe relies on earlier-layer heads, CoRe-R maintains strong accuracy even under aggressive layer pruning, achieving a better accuracy–efficiency tradeoff.
>
> In the revision, we extend this analysis and additionally report overlap statistics for the top-32 and top-64 heads across all methods (Appendix B.1).
>
> ### 4. There are missing baselines for the layer pruning experiments. Since the retrieval heads (i.e., NIAH-R) in Wu et al., 2024 also concentrate in the middle layers, I do not clearly see how layer pruning can only be applied to CoRe-R to improve efficiency, but not other attention-based re-rankers that use a sparse set of attention heads. Thus, the following baselines should be added for fairer evaluation of the efficiency benefit of the proposed method: QR-R (prune) and NIAH-R (prune).
>
> We thank the reviewer for this very helpful suggestion. We have added QR-R (prune) and NIAH-R (prune) as baselines for both Mistral-7B and Llama-3.1-8B, and the updated results are shown in Figure 14 (Appendix C.4) of the revision.
>
> It is worth noting that, to the best of our knowledge, prior work on attention-based methods (including ICR, QR-R, and NIAH-R) has not explored the idea of pruning model layers for efficiency. Our study is therefore the first to systematically examine layer pruning in this setting, and the additional baselines requested by the reviewer actually reinforce this contribution.
>
> Our new experiments show that several sparse-head methods can indeed benefit from pruning, which is an interesting and previously unreported observation. However, across both Mistral and Llama, CoRe-R is consistently **the most robust**: it retains near-identical accuracy up to 50\% pruning, while QR-R begins degrading much earlier (around 40\%) and NIAH-R drops sharply beyond 50\%. This aligns with our head-location analysis: CoRe heads lie in earlier/middle layers, whereas QR and NIAH heads rely more on the late layers that are pruned away.
>
> ### 5. Why is contextual calibration only used for the re-ranking process, but not the head detection process of CoRe?
>
> When comparing different documents during the re-ranking process, intrinsic bias (e.g. document length, punctuation) can occur, and contextual calibration helps to cancel out these biases. However, during the head detection process, all attention heads are equally evaluated on the same set of documents so there is no bias issue. In addition, preserving the intrinsic bias eliminates low-quality heads, leaving high-quality non-biased heads as retrieval heads. We have added this discussion to the revision (lines 190-192).
>
> ### 6. Figure 4: The head distribution in Mistral 7B is very different from those in the other models. Do you have any insights into this?
>
> The difference in Mistral's head distribution is due to lower temperature compared to other models. We set the temperature for Mistral to be $t=0.001$ which is a lot lower than other models where $t=0.1$. The low temperature increases the penalties on hard negatives and raises the gap between the scores of high-quality and low-quality heads. This sharpens the head score distribution, nevertheless, the trend in Mistral is the same with other models: most heads in the mid layer have the highest scores. We have added this explanation to the revision (lines 407-408). We provided in Appendix B.2 (also appeared in the original manuscript) the head distribution in Mistral with higher temperatures, and we observe that Mistral with $t=0.1$ has similar distribution with other models that use the same temperature.

---

### Author Response · Authors · 2025-11-21
**General Response**

We thank all reviewers for taking the time to review our paper and providing constructive comments and suggestions. Below we provide clarification to address common concerns from the reviewers.

## Clarification on the methodology.
Reviewers ivKj and 3Gtp raised similar concerns regarding (1) whether CoRe normalizes the document-side attention scores while ignoring the instruction, and (2) whether CoRe’s contrastive scoring is effectively equivalent to the softmax, which already encodes relative weighting. We believe these concerns stem from our original phrasing, specifically the statement on line 183 in the original manuscript that certain tokens were "ignored in attention score computation.'' We have revised the wording and updated the mathematical equations (still equivalent to previous version) to avoid this misunderstanding. We sincerely apologize for the lack of clarity and address both reviewers’ concerns below.

* **Clarification on how instruction tokens are treated.** All methods (ICR, QR, and CoRe) rely on the *same* token-level attention scores produced by the model’s native softmax, which already normalizes over the *entire* context, including the instruction. The instruction tokens are excluded *only* during the aggregation step that constructs *document-level* scores, since the instruction does not belong to any document. As a result, QR and CoRe begin from identical document-level attention scores.

* **Clarification on the role of softmax vs. CoRe’s contrastive scoring.**  The model’s internal softmax normalizes attention on a *token-level*, whereas CoRe introduces a contrastive scoring metric that operates on a *document-level*. Because instruction tokens are removed during document-level aggregation, the aggregated document scores are not necessarily normalized. This means a head may appear “good’’ under QR’s absolute scoring simply because it assigns a large score to the positive document, even if it assigns an *even larger* score to semantically similar hard negatives.

    CoRe explicitly incorporates hard negatives and applies a contrastive objective inspired by InfoNCE to penalize heads that attend strongly to misleading negatives. This yields a genuinely *relative* and discriminative assessment of a head’s selectivity, fundamentally different from QR’s absolute scoring, and explains CoRe’s improved robustness.

---

### Author Response · Authors · 2025-11-21
**General Response**

## Summary of additional experiments.
We thank the reviewers for their constructive suggestions. In response, we conducted an extensive set of new experiments, and the resulting findings further strengthen and reinforce the contributions of the work.
* **Performance at long context (Reviewer 3Gtp).** We extended the reranking experiments to top-60, top-80 and top-100 on BEIR (results reported in Figure 4 and Appendix C.6). Relative to the top-40 evaluations, CoRe-R continues to achieve the highest nDCG@10 scores and shows *even larger* gains over the baselines, further widening the performance gap (see Figure 4). The intuition is that with long context, the candidate set inevitably contains more semantically similar hard negatives. CoRe’s contrastive scoring explicitly penalizes non-selective heads by comparing positives against their associated negatives, making it robust to noise and dilution effects that arise when attention mass is spread across long contexts. Therefore, CoRe is particularly effective in long context settings where hard negatives become more prevalent and the ranking task is inherently more challenging.
*  **Generalizability to multi-hop tasks (Reviewer 3Gtp).** In addition to the multi-hop dataset HotpotQA, we ran experiments on two new multi-hop datasets MuSiQue and CLIPPER (results reported in Figure 3 and Appendix C.5). On the new multi-hop tasks, CoRe-R still delivers the best reranking performance compared to the baselines, indicating the applicability of CoRe in both single-hop and multi-hop retrieval tasks.
* **Sensitivity to the detection dataset (Reviewer fEwK).** To evaluate the robustness of CoRe head selection, we repeated the detection procedure using a different dataset, MSMARCO. The resulting CoRe heads are nearly identical to those detected from NQ: for both Mistral and Llama, 7 out of the 8 selected heads overlap, and the one different head comes close within the top10 heads. We then performed the reranking experiments using these MSMARCO-derived heads (results reported in Appendix C.7). With 7 out of 8 heads overlapping, the reranking performance closely matches the results obtained using NQ, demonstrating that CoRe head selection is highly stable and does not depend heavily on the detection dataset.
* **Optimal number of heads (Reviewers fEwK and ivKj).** We provide additional analysis to examine how the optimal number of CoRe heads varies across datasets. We replicated the “number of heads’’ experiment from Figure 2 on three additional datasets using Mistral and Llama (results reported in Appendix C.8). Across all new datasets, we observe the same consistent trend: reranking performance peaks within the first 10 CoRe heads. This demonstrates that CoRe does not require dataset-specific tuning of the head count and that a small, fixed number of heads suffices in practice.
* **Effect of the temperature (Reviewer ivKj).** We performed ablation study on the temperature $t$ (results reported in Section 5.5). Since $t$ directly controls the sharpness of the contrastive scoring metric, different temperatures can change the *shape* of the score distribution and may affect which heads appear in the top-ranked set. However, despite these shifts in absolute values, we observe that the *most informative CoRe heads remain consistent* across temperatures, and their layer distribution is highly stable. While a small amount of temperature tuning is required, we emphasize that head detection is an extremely efficient, one-time procedure per model: detecting high-quality CoRe heads for a 7B–8B model takes less than an hour on H100 96GB GPU, and this process can be done in parallel for multiple LLMs. Once selected, the same CoRe heads generalize reliably across cross-domain tasks, multi-hop retrieval, and long-context settings. This demonstrates that CoRe’s performance is not sensitive to temperature in practice, and the detected heads remain robust across applications.
* **Accuracy-efficiency tradeoff via layer pruning (Reviewer ipmp).** We added pruning experiments for the baselines NIAH-R and QR-R (results in Appendix C.4). We observe that both NIAH and QR rely on a substantial number of mid-layer heads. Consequently, their performance remains stable under light pruning (30–40\%).  However, both baselines begin to degrade significantly once pruning exceeds 40\%, whereas CoRe-R maintains near-identical accuracy up to 50\% pruning. This difference is intuitive: all CoRe heads lie in early– to mid–layers, while NIAH and QR additionally rely on several late-layer heads (layer 20 and beyond), which are removed earlier in the pruning schedule. Overall, CoRe-R not only achieves the highest reranking accuracy but also demonstrates the strongest accuracy-efficiency tradeoff, benefiting directly from the structurally favorable distribution of its selected heads.

---

### Author Response · Authors · 2025-12-03
**Summary of the Rebuttal**

We thank the reviewers for their helpful comments and suggestions. We are glad to see that the reviewers appreciate the following aspects of our work:
* CoRe is a new and novel method (ipmp) that is effective (3Gtp), efficient and practical for real-world applications (fEwK).
* We cover an important system consideration (ivKj) that allows advancement in LLM interpretability and opens avenues for optimized architectures (Reviewer fEwK).
* We conduct comprehensive empirical study across four open-source LLMs (3Gtp, ipmp) with extension to multilingual settings that were not previously studied (ivKj).
* The results and analysis are strong (ivKj) with consistent improvements over baselines (fEwK).

During the rebuttal period, we have addressed **all** reviewer comments and substantially strengthened the paper. Key changes include:

* **Deeper analysis and insights of empirical results (ipmp, fEwK, ivKj)**

Several reviewers asked why CoRe-R shows **large improvements in some settings** (e.g., Quora with Mistral-7B) but smaller gains for Llama and Phi.
First, we now show that **all observed improvements are statistically significant**, directly addressing concerns about effect magnitude. Second, we provide detailed analysis explaining the larger gaps: the Quora dataset contains many **hard negatives**, and Mistral-7B exhibits many **ambiguous heads**, i.e., heads that attend strongly to both positives and hard negatives. CoRe’s contrastive scoring explicitly penalizes such ambiguity, which aligns perfectly with these settings. This explains the large gains and validates that **contrastive head selection is particularly effective in semantically dense or near-duplicate domains**.

* **Clarified methodology and contribution (ivKj, 3Gtp)**

We revised the description of the attention score calculation and updated the mathematical equations to clearly distinguish our contrastive scoring from the LLM’s internal softmax, resolving earlier misunderstandings about novelty.

* **Expanded retrieval-head analysis (ipmp, 3Gtp)**

We added detailed comparisons of top-8, top-32, and top-64 heads. The analyses show minimal to no overlap between CoRe and baseline (NIAH, QR) heads. CoRe heads consistently appear in earlier layers and carry strong contrastive signal, explaining CoRe-R’s superior accuracy–efficiency tradeoff.

 * **Optimal number of heads (fEwK, ivKj)**

To address questions about the optimal number of heads, we expanded this experiment to **three additional datasets** and **two LLMs**. The new results consistently confirm the earlier finding (Fig. 2): reranking performance reliably **peaks within the first 10 CoRe heads**, demonstrating that CoRe does **not** require tuning of the head count.

* **Hard negatives and temperature ablations (fEwK, ivKj)**

Using an alternative detection dataset, we find CoRe heads to be stable and dataset-independent. Temperature affects the sharpness of the contrastive scoring metric but requires only light, one-time tuning per model.

* **Generalizability to multihop and long context settings (3Gtp)**

New experiments on multihop datasets and extended long-context settings show that CoRe-R **consistently outperforms** baselines, with the improvement gap **widening** as context length increases.

* **Baselines with layer pruning (ipmp)**

We added layer-pruning experiments for QR-R and NIAH-R. Their performance remains stable only up to ~40\% pruning, after which accuracy degrades. In contrast, CoRe-R maintains nearly identical performance up to **50\% pruning**, achieving the strongest accuracy–efficiency tradeoff.

We also addressed remaining clarification questions, including contextual calibration and head-distribution behavior (ipmp) and hard-negative criteria (fEwK).

**Overall, we have thoroughly addressed every reviewer comment and significantly strengthened both the empirical and methodological components of the paper.** The expanded experiments and analyses consistently validate the effectiveness, robustness, and practical relevance of CoRe-R. We thank the reviewers for their constructive feedback.

---

### Meta-Review · Area_Chair_vEpX · 2025-12-21

**Summary:**

This paper proposes a new attention-based re-ranker that identifies contrastive retrieval heads in LLMs to improve document re-ranking by focusing on relevant information. Empirical results demonstrate that this method outperforms existing baselines. The authors have revised lots of parts of the paper, but these are still some concerns should be further revised.

**Reviewer Concerns:**

The authors have made progress in addressing several of the reviewers' concerns:
1. The method section has been updated, providing clearer explanations.
2. Some issues related to experimental settings have been addressed.

However, there are still a few unresolved concerns:
1. The performance of the contrastive retrieval head re-ranker is relatively weak compared to attention-based re-ranker baselines.
2. The novelty of the paper remains limited. The proposed method introduces only incremental changes compared to prior works.
3. A deeper theoretical analysis of how these differences contribute to performance improvements, as well as a clearer comparison with previous methods, would help strengthen the paper's contribution.

**Reviewer Scores:**

No score may be changes.

---

### Decision · Program_Chairs · 2026-01-26

Reject